# Updated vaccine protects against SARS-CoV-2 variants including Omicron (B.1.1.529) and prevents transmission in hamsters

Sapna Sharma [1], Thomas Vercruysse [2], Lorena Sanchez-Felipe [1], Winnie Kerstens[2], Madina Rasulova [2], Lindsey Bervoets[1], Carolien De Keyzer[1], Rana Abdelnabi [1], Caroline S. Foo [1], Viktor Lemmens [1], Dominique Van Looveren [2], Piet Maes [3], Guy Baele [4], Birgit Weynand[5], Philippe Lemey [4], Johan Neyts [1,6], Hendrik Jan Thibaut [2,7] & Kai Dallmeier [1,7] ✉

Current COVID-19 vaccines are based on prototypic spike sequences from ancestral 2019 SARS-CoV-2 strains. However, the ongoing pandemic is fueled by variants of concern (VOC) escaping vaccine-mediated protection. Here we demonstrate how immunization in hamsters using prototypic spike expressed from yellow fever 17D (YF17D) as vector blocks ancestral virus (B lineage) and VOC Alpha (B.1.1.7) yet fails to fully protect from Beta (B.1.351). However, the same YF17D vectored vaccine candidate with an evolved antigen induced considerably improved neutralizing antibody responses against VOCs Beta, Gamma (P.1) and the recently predominant Omicron (B.1.1.529), while maintaining immunogenicity against ancestral virus and VOC Delta (B.1.617.2). Thus vaccinated animals resisted challenge by all VOCs, including vigorous high titre exposure to the most difficult to cover Beta, Delta and Omicron variants, eliminating detectable virus and markedly improving lung pathology. Finally, vaccinated hamsters did not transmit Delta variant to non-vaccinated cage mates. Overall, our data illustrate how current first-generation COVID-19 vaccines may need to be updated to maintain efficacy against emerging VOCs and their spread at community level.

Severe Acute Respiratory Syndrome Corona Virus 2 (SARS-CoV-2) emerged as a zoonosis likely from a limited number of spill-over events into the human population[1]. Nevertheless, the ongoing COVID-19 pandemic is driven by variants that evolved during subsequent large-

scale human-to-human transmission. Particularly mutations within the viral spike protein are under continuous surveillance considering its role in viral pathogenesis and as target of virus-neutralizing antibodies (nAb). Following early diversification, the D614G SARS-CoV-2 variant

[1]KU Leuven Department of Microbiology, Immunology and Transplantation, Rega Institute, Laboratory of Virology, Molecular Vaccinology and Vaccine Discovery, BE-3000 Leuven, Belgium. [2]KU Leuven Department of Microbiology, Immunology and Transplantation, Rega Institute, Laboratory of Virology and Chemotherapy, Translational Platform Virology and Chemotherapy, BE-3000 Leuven, Belgium. [3]KU Leuven Department of Microbiology, Immunology and Transplantation, Rega Institute, Laboratory of Clinical and Epidemiological Virology, Zoonotic Infectious Diseases Unit, BE-3000 Leuven, Belgium. [4]KU Leuven Department of Microbiology, Immunology and Transplantation, Rega Institute, Laboratory of Clinical and Epidemiological Virology, Evolutionary and Computational Virology, BE-3000 Leuven, Belgium. [5]KU Leuven Department of Imaging and Pathology, Translational Cell and Tissue Research, BE-3000 Leuven, Belgium. [6]Global Virus Network (GVN), Baltimore, MD, USA. [7]These authors contributed equally: Hendrik Jan Thibaut, Kai Dallmeier. ✉e-mail: kai.dallmeier@kuleuven.be

(B.1 lineage) became dominant in March 2020. Late 2020, Variants of Concern (VOC) emerged with increased transmissibility, potentially increased virulence and escape from naturally acquired and vaccine-induced immunity[2]. Compared to prototypic (Wuhan) or early European D614G (B.1) lineages of SARS-CoV-2, the first four recognized VOCs harbor each a limited set of partially convergent, partially unique spike mutations, namely VOC Alpha (B.1.1.7; N501Y D614G), Beta (B.1.351; K417N E484K N501Y D614G), Gamma (P.1; K417T E484K N501Y D614G), and Delta (B.1.617.2; K417T L452R T478K D614G P681R)[3]. N501Y was first detected in VOC Alpha and has been linked to an enhanced transmissibility due to an increased affinity for the human[4,5] and mouse[6] ACE2 receptor. Subsequent emergence of E484K within this lineage hampers the activity of nAb suggestive for immune escape[7–9]. Likewise, a combination of K417N and E484K[10] may explain a marked reduction in vaccine efficacy (VE) of some vaccines such as ChAdOx1 nCoV-19 (AstraZeneca, Vaxzevria) in clinical trials in South Africa during high prevalence of VOC Beta[11]. Similarly, sera from vaccinees immunized with first-generation mRNA (Pfizer-BioNTech, Comirnaty; Moderna, mRNA-1273) or nanoparticle subunit vaccines (Novavax) showed a substantial drop in neutralizing capacity for VOC Beta[12]. Furthermore, VOC Gamma harboring K417T and E484K emerged in regions of Brazil with high seroprevalence, despite naturally acquired immunity against prototypic SARS-CoV-2[13]. VOC Delta was first identified in October 2020 in India[14] and became the predominant SARS-CoV-2 lineage worldwide in 2021, driven by a substantially increased transmissibility[15]. In late November 2021, a new VOC Omicron (B.1.1.529) was discovered in southern Africa[16] and is since then spreading globally; displacing other strains at unprecedented speed. Omicron carries the by far the largest number (>32) of mutations, deletions, and insertions in its spike protein described to date[17,18], including a combination of substitutions previously linked to increased human-to-human transmission (N501Y D614G P681H) as well as escape from antibody-mediated immunity (K417N E484A) acquired by natural exposure or elicited by current vaccines[19,20]. While the intrinsic pathogenic potential of Omicron remains uncertain[21], its antigenic divergence leads to a loss of activity of most therapeutic monoclonal antibodies[22] and failure of current first-generation vaccines to protect from infection[23,24]. Maintenance of even minimal cross-protective nAb levels requires repeated booster dosing[24–26].

Currently licensed COVID-19 vaccines and still most vaccine candidates in advanced clinical development are based on antigen sequences of early SARS-CoV-2 isolates that emerged in 2019[27]. Likewise, we reported on a YF17D-vectored SARS-CoV-2 vaccine candidate using prototypic spike as vaccine antigen (YF-S0; S0) that had a promising preclinical efficacy against homologous challenge[28]. Considering the rapidly changing antigenicity of SARS-CoV-2, it remains challenging to develop a universal vaccine that follows the evolution of future SARS-CoV-2 variants that carry like VOC Omicron increasingly complex combinations of old and new mutations[22,29,30] responsible for both nAb escape (e.g., E484K/A)[10] and enhanced transmission (e.g., N501Y; P681R/H)[31].

Here we show to what extent VE of S0, and hence first-generation spike vaccines in general, may decline when trialed against VOC in a stringent hamster model[32]. To address whether updated vaccines could cover a wider VOC spectrum, we designed a second-generation vaccine candidate (YF-S0*) by (i) modifying its antigen sequence to resemble that of VOC Gamma[33], in combination with (ii) a further stabilized protein conformation[34]. This new S0* vaccine candidate provides protection against all currently recognized VOCs Alpha, Beta, Gamma, and Delta as well as most recent highly diverse Omicron. No replicating virus was detected in lungs of vaccinated hamsters, whereas high virus loads were found in lungs of non-vaccinated controls. A marked increase in VOC-specific nAb levels after vaccination with S0* serve as proxy for an improved VE. Finally, hamsters vaccinated with S0* no longer transmit the virus to non-vaccinated sentinels during close contact, even under conditions of prolonged co-housing and exposure to a high infectious dose of VOC Delta. Moreover, our findings clearly demonstrate to what large extent current first-generation COVID-19 vaccines are outdated, lose VE and can be expected to fail against recent and future VOCs that fuel the ongoing pandemic. Based on the data presented here, YF-S0* may be considered a potent second-generation vaccine candidate worth to progress to clinical trials.

## Results

### No change regarding VOC Alpha, yet markedly reduced efficacy of first-generation spike vaccine against VOC Beta

To assess VE of prototypic spike antigen against VOCs, hamsters were vaccinated using a previously validated rapid immunization scheme[28], i.e., twice with each $10^4$ PFU of YF-S0 (S0) at day 0 and 7 via the intraperitoneal route, or sham (Fig. 1A). Serological analysis at day 21 confirmed that 30/32 (94%) vaccinated hamsters had seroconverted to high levels of nAbs against prototypic SARS-CoV-2 with geometric mean titre (GMT) of 2.3 $\log_{10}$ (95% CI 2.0–2.6) (Fig. 1B). Next, animals were challenged intranasally with $1 \times 10^3$ TCID50 of either prototypic SARS-CoV-2, VOC Alpha or Beta as established and characterized before in the hamster model[32]. At day 4 after infection (4 dpi), viral replication was determined in lung tissue by qPCR and virus titration (Fig. 1C, D). In line with what was originally described for S0[28], a marked reduction in viral RNA and in particular of infectious virus loads down to undetectable levels (up to 6 $\log_{10}$ reduction) was observed in the majority of animals challenged with either prototypic SARS-CoV-2 (8/10; 86% VE) or VOC Alpha (9/10; 88% VE). In those animals (2/10 and 1/10, respectively) that were not completely protected, virus loads were at least 100 times lower than in infected sham controls. By contrast and despite full immunization, S0 vaccination proved to be less effective against VOC Beta, with only 4/12 hamsters without detectable infectious virus (60% VE). Nonetheless, in the remaining 8/12 animals with breakthrough infection by VOC Beta, viral replication was tempered as vaccination still resulted in a 10–100-fold reduction in infectious virus titres relative to sham. In general, viral RNA levels and infectious virus titres followed the same trend, though individual reductions in viral RNA levels (Fig. 1C, D) did not always result in a similar reduction in virus titres. This is not unexpected considering (i) the higher sensitivity and dynamic range of qPCR, and likewise (ii) the general observation that viral RNA detected by qPCR may represent residues originating from cellular debris rather than viral genomes actively involved in an ongoing productive infection.

Logistic regression was used to define immune correlates of protection[35]. This analysis confirmed that comparable nAb levels were required for protection against prototypic SARS-CoV-2 (1.5 $\log_{10}$ for 50% and 2.9 $\log_{10}$ for 90% protection) and VOC Alpha (1.2 $\log_{10}$ for 50% and 2.5 $\log_{10}$ for 90% protection) (Fig. 1E). Intriguingly, for VOC Beta a markedly (up to 25x) higher nAb threshold (2.6 $\log_{10}$) was required for 50% protection. Importantly, no 90% protective nAb threshold could be defined anymore for VOC Beta infection, considering the high number of S0-vaccinated animals with viral breakthrough (>$10^2$ TCID50/100 mg lung tissue)[35]. Overall, these data suggest that first-generation vaccines employing prototypic spike as antigen suffer from a markedly reduced efficacy against emerging SARS-CoV-2 variants, such as VOC Beta.

### Updated spike antigen offers protection against full range of VOCs Alpha, Beta, Gamma, and Delta

Although prototype S0 showed induction of high nAb levels against prototypic SARS-CoV-2 (Fig. 1B) and protective immunity against prototypic SARS-CoV-2 and VOC Alpha (Fig. 1C–E), prototypic spike antigen failed to induce consistent nAb responses against more recent VOCs (Fig. 2A). Thereby YF-S0 vaccination resulted in particularly poor seroconversion and low nAb titres against VOC Beta (seroconversion

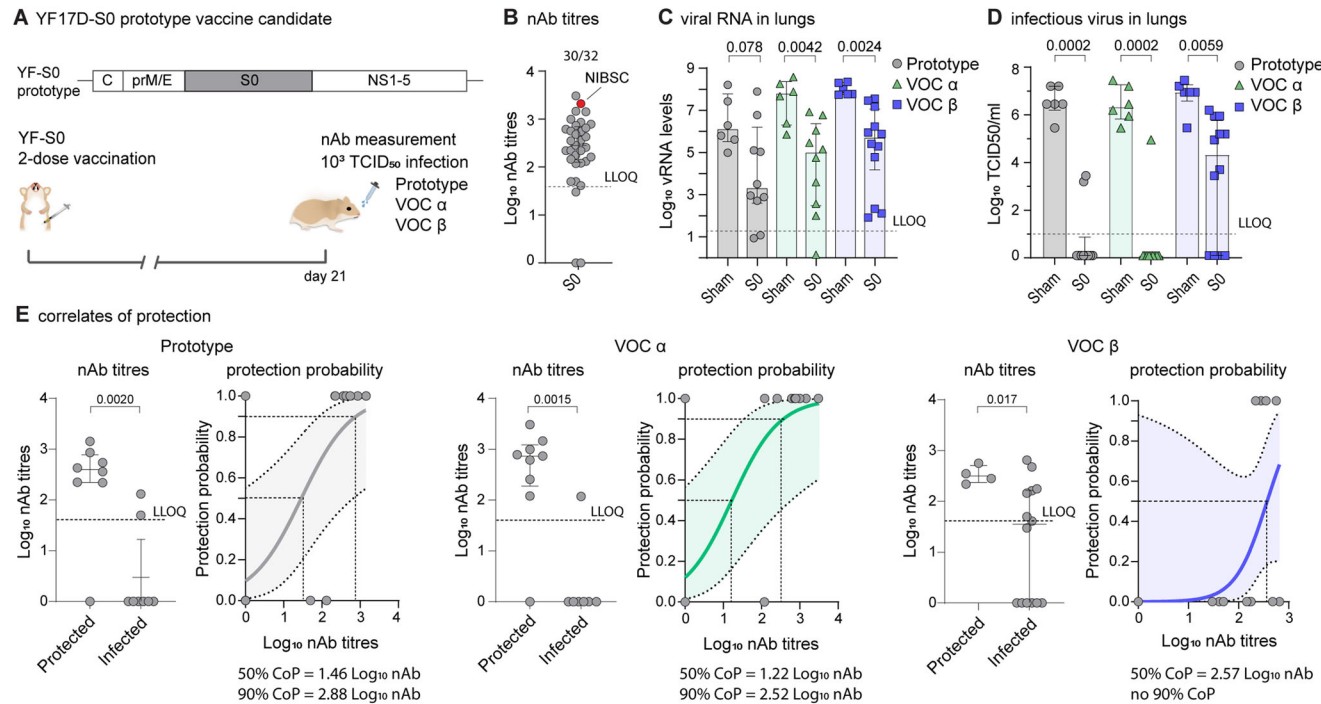

**Fig. 1 | Immunogenicity and protective efficacy of first-generation Spike vaccine against prototype SARS-CoV-2 and VOCs Alpha and Beta. A** Vaccination scheme with prototypic YF17D-based vaccine candidate YF-S0 (S0). Syrian hamsters ($n = 32$ vaccinated and $n = 18$ sham vaccinated) were immunized twice intraperitoneally with $10^4$ PFU of S0 on day 0 and 7 and inoculated intranasally on day 21 with $10^3$ median tissue-culture infectious dose ($TCID_{50}$) of either prototype SARS-CoV-2 (gray circles, $n = 10$ YF-S0 and $n = 6$ sham), VOC Alpha (green triangles, $n = 10$ YF-S0 and $n = 6$ sham), or VOC Beta (blue squares, $n = 12$ YF-S0 and $n = 6$ sham). **B** nAb titres against prototypic spike (D614G) pseudotyped virus on day 21 after vaccination, 30/32 indicates the seroconversion rates. Red datapoint indicates the

NIBSC 20/130 human reference sample included as benchmark. **C**, **D** Viral loads in hamster lungs 4 days after infection quantified by quantitative RT-PCR (**C**) and virus titration (**D**). **E** correlates of protection against prototype SARS-CoV-2, VOC Alpha, and VOC Beta (analysis per group as in **D**). Logistic regression model to calculate nAb titres correlating with 50 and 90% probability for protection. Protected was defined by a viral load $<10^2$ TCID50/ml lung tissue and infected by a viral load $>10^2$ TCID50/ml lung tissue (van der Lubben et al., 2021). Shaded areas indicate 95% CI. LLOQ is lower limit of quantification. Bar graphs denote median ± IQR. Differences between groups were analyzed using nonparametric Kruskal–Wallis test uncorrected for ties.

rate 15/32; GMT 1.0 $\log_{10}$, 95% CI of 0.6–1.3) and Gamma (19/32; GMT 1.3 $\log_{10}$, 95% CI 0.9–1.8). Intriguingly, human convalescent plasma used as benchmark (WHO standard NIBSC 20/130) originating from 2020 prior to the surge of VOC (Fig. 2A, B) showed a similar loss of activity against VOC Beta, in line with what we observed in our hamster sera (Fig. 2A).

It is not clear if the full spectrum of antigenic variability of current VOCs and emerging variants can be covered by a COVID-19 vaccine that is based on a single antigen[36,37]. In an attempt to generate a more universal variant-proof SARS-CoV-2 vaccine (YF-S0*, S0*), we adapted the spike sequence in our original YF-S0 construct to include all 12 amino acid changes originally found in VOC Gamma (not a priori limited to known immunogenic sites)[33], plus three extra proline residues (A892P, A942P, and V987P) to stabilize spike in a conformation favorable for immunogenicity[34,38] (Fig. 2C). Thus, in combination with three amino acid changes (RRAR > AAAA) ablating S1/S2 cleavage[28], in total 18/1260 amino acid residues (1.4% of prefusion spike, disregarding N-terminal signal peptide) were changed in the new antigen (S0*) compared to the prototypic spike sequence.

YF-S0* proved to be highly immunogenic against prototypic SARS-CoV-2, with nAb levels reaching GMT of 2.2 $\log_{10}$ (95% CI 1.8–2.6) and a seroconversion rate of 21/24 (Fig. 2D), comparable to original YF-S0 (GMT 2.3 $\log_{10}$, 95% CI 2.0–2.6; 30/32 seroconversion rate) (Fig. 2A, Fig. S1). Also, for both constructs, seroconversion rates and nAb levels against VOC Delta were equally high (YF-S0: 30/32; GMT 2.0 $\log_{10}$, 95% CI 1.7–2.2; YF-S0*: 22/24; GMT 2.0 $\log_{10}$, 95% CI 1.6–2.3). Notably, for YF-S0*, nAb levels and seroconversion rates against previously poorly covered VOC Beta (GMT 2.9 $\log_{10}$, 95% CI 2.6–3.2; seroconversion rate

23/24) and Gamma (GMT 3.0 $\log_{10}$, 95% CI 2.8–3.2; seroconversion rate 24/24) were markedly increased (by 50 to 80-fold for GMT; 1.7 to 2-times more frequent seroconversion) (Fig. 2A, D and Fig. S1).

S0*-vaccinated animals were subsequently challenged with each $10^3$ TCID50 of either of the four VOCs Alpha, Beta, Gamma or Delta, and sacrificed 4 dpi for assessment of viral loads in the lung (Fig. 2E, F) and associated lung pathology (Fig. 2G, H). In S0*-vaccinated hamsters, viral RNA loads were uniformly reduced compared to matched sham controls by up to ~5–6 $\log_{10}$ (VOC Alpha, Beta and Gamma) or by at least ~3 $\log_{10}$ (VOC Delta) (Fig. 2E). Importantly, no infectious virus could be detected anymore (~6 $\log_{10}$ reduction) in any of the animals vaccinated with S0*, irrespective of which VOC they had been exposed to (Fig. 2F), confirming 100% VE conferred by S0* against all four VOCs.

Protection from infection also translated in a markedly reduced lung pathology (Fig. 2G, H). Non-vaccinated sham animals developed characteristic signs of bronchopneumonia with perivascular and peribronchial infiltrations, edema and consolidation of lung tissues[32,39]. In contrast, lungs of S0*-vaccinated hamsters remained markedly less affected with a clear reduction in overall histological scores, irrespectively of the VOC used (Fig. 2G, H). In conclusion, second-generation YF-S0* expressing an updated S0* antigen induced consistently high levels of broadly neutralizing antibodies (Fig. 2D) which translated into efficient protection from lower respiratory tract infection and COVID-19-like pathology by a large spectrum of VOCs (Fig. 2G, H). VE of S0* covered VOCs Beta and Gamma, i.e., variants harboring key mutations K417N/T and E484K escaping original spike-specific nAb activity (Fig. 2B), and may therefore also offer better protection against other emerging

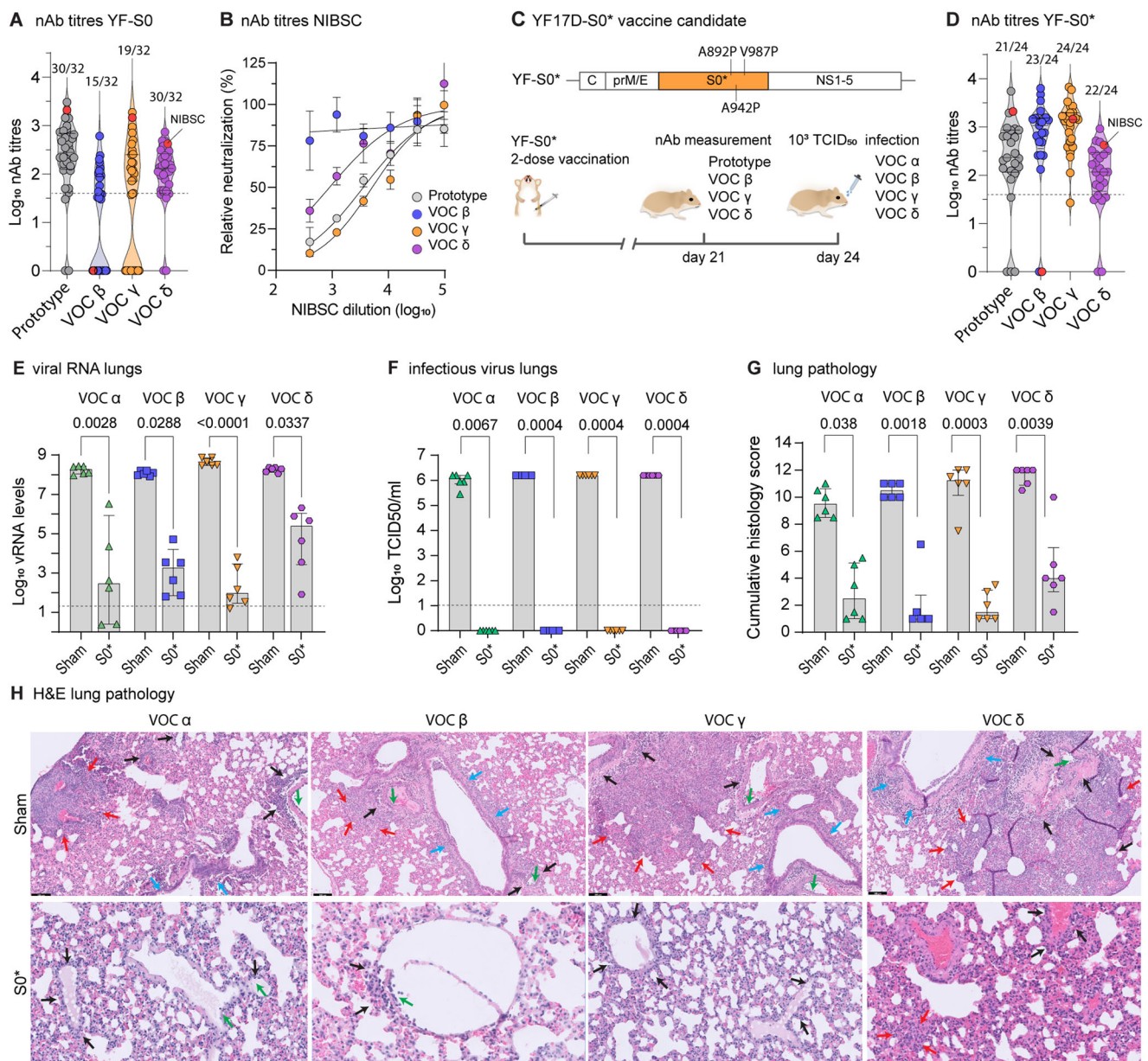

**Fig. 2 | A vaccine based on updated spike antigen S0* offers protection against four VOCs. A** nAb titres against prototypic (gray circles), VOC Beta (blue circles), VOC Gamma (orange circles) and VOC Delta (purple circles) spike-pseudotyped virus on day 21 after vaccination with prototype YF-S0. Numbers above individual violin plots indicate animals seroconverted. Red datapoint indicates the NIBSC 20/130 human reference sample. NAb titres against prototypic spike-pseudotyped virus (gray circles) are from Fig. 1B. **B** Neutralization curves for NIBSC 20/130 human reference sample against same set of pseudotyped viruses. **C** Schematic of the updated YF-S0* (S0*) vaccine candidate based on VOC Gamma, plus three extra stabilizing proline residues. Vaccination scheme with YF-S0*. Syrian hamsters were immunized twice intraperitoneally with $10^4$ PFU of S0* (n = 24 vaccinated and n = 24 sham vaccinated) on day 0 and 7 and inoculated intranasally on day 24 with $10^3$ median tissue-culture infectious dose (TCID50) of either VOC Alpha (green triangles), VOC Beta (blue squares), VOC Gamma (orange triangles), and VOC Delta (purple circles). **D** nAb titres against prototypic, VOC Beta, VOC Gamma, and VOC Delta spike-pseudotyped virus on day 21 after vaccination with YF-S0*. Numbers above individual violin plots indicate animals seroconverted. Red datapoint indicates the NIBSC 20/130 human reference sample. **E, F** Viral loads in hamster lungs 4 days after infection (n = 6 each YF-S0* or sham vaccinated per variant) quantified by quantitative PCR with reverse transcription (RT-qPCR) (**E**) and virus titration (**F**). **G** cumulative lung pathology scores from H&E-stained slides of lungs for signs of damage. **H** Representative H&E-stained images of sham- or S0*-vaccinated hamster lungs after challenge. Perivascular inflammation (black arrows) with focal endothelialitis (green arrows); peribronchial inflammation (blue arrows); patches of bronchopneumonia (red arrows); scale bar upper row images 100 μm, lower row images 50 μm. Bar graphs denote median ± IQR. Dashed line represents the lower limit of quantification. Differences between groups were analyzed using nonparametric Kruskal–Wallis test uncorrected for ties.

variants with similar signatures such as Variant of Interest (VOI) Mu (E484K), or more recently VOC Omicron.

## Protection against high-dose VOCs Beta challenge
VOC Beta appeared to be the most difficult variant to cover by prototypic YF-S0 vaccination (Fig. 2A), even by using a two-dose vaccination schedule for immunization (twice $10^4$ PFU of YF-S0, 7 days

apart) in combination with a moderate $10^3$ TCID50 dose of VOC Beta for intranasal challenge (Fig. 1D). To explore the limits of its potency, the immunogenicity and VE of YF-S0* was further assessed under more stringent conditions; by (i) comparing the previous two-dose to a single-dose vaccination schedule followed (ii) by a more vigorous challenge using a 100-fold higher $10^5$ TCID50 dose of VOC Beta for infection (Fig. 3A). 100% seroconversion against VOC Beta was

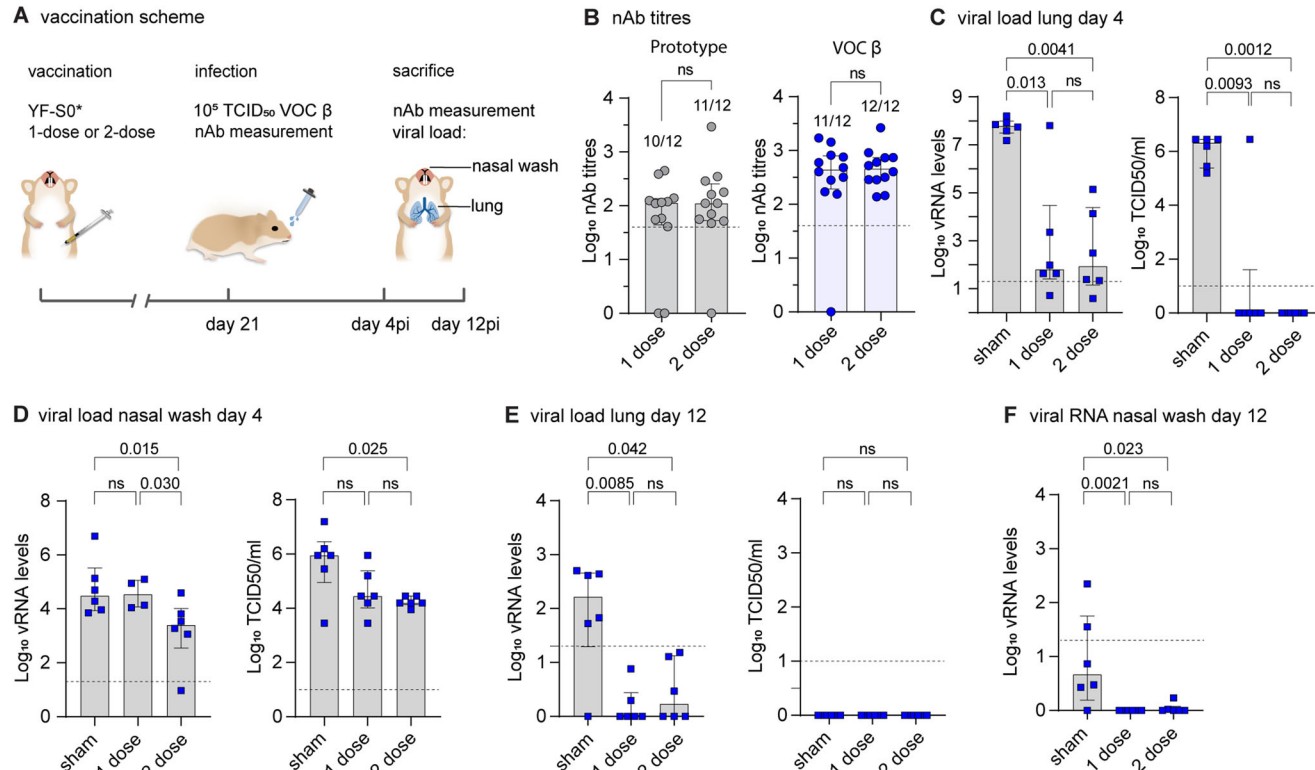

**Fig. 3 | Protection by YF-S0\* against vigorous high-dose VOC Beta challenge.**
**A** Vaccination scheme with updated YF-S0\*. Syrian hamsters were immunized either once ($n = 12$) or twice ($n = 12$) intraperitoneally with $10^4$ PFU of YF-S0\* and $n = 12$ hamsters were sham vaccinated and inoculated intranasally on day 21 with $10^5$ median tissue-culture infectious dose (TCID50) of VOC Beta. Animals were sacrificed on either day 4 ($n = 18$) or day 12 ($n = 18$) after infection (each timepoint $n = 6$ each per group; once or twice YF-S0\* vaccinated, and sham). **B** nAb titres against prototypic (gray circles) and VOC Beta (blue circles) spike-pseudotyped virus on day 21 after vaccination with prototype YF-S0\*. Numbers above bars indicate animals seroconverted. **C–F** Viral loads in hamster lungs (**C**, **E**) and nasal wash (**D**, **F**) 4 (**C**, **D**), or 12 (**E**, **F**) days after infection quantified by quantitative PCR with reverse transcription (RT-qPCR) and virus titration (blue squares). RNA extracts from 1-dose sample from 2/6 animals were not available to analyze viral RNA levels (**D**). Bar graphs denote median ± IQR. Dashed line represents the lower limit of quantification. Differences between groups were analyzed using nonparametric two-tailed Mann–Whitney (**B**) or Kruskal–Wallis test uncorrected for ties (**C–F**).

observed in the two-dose group (12/12; $\log_{10}$ GMT 2.7 IQR 2.5–2.9) while in the single-dose group one hamster did not seroconvert against VOC Beta (seroconversion rate 11/12; $\log_{10}$ GMT 2.5 IQR 2.3–2.9) (Fig. 3B, VOC Beta). Likewise, in total 21/24 animals (10/12 from single-dose; $\log_{10}$ GMT 1.7 IQR 1.6–2.1 and 11/12 from two-dose; $\log_{10}$ GMT 2.0 IQR 1.7–2.4) had also high levels of nAb specific for prototypic spike (Fig. 3B, prototype). Overall, a second dose after 7 days helped to close the remaining gaps in immunization and reduced the chance of primary vaccination failure from <13% (3/24) to <5% (1/24). It is reasonable to assume that prolonging the interval between the two doses will lead to a further improved prime-boost regimen (see Methods).

Following VOC Beta challenge all sham animals developed consistently high viral loads at peak of infection, both in lungs (Fig. 3C) as well as in nasal washes day 4 p.i. (Fig. 3D). Notably, as described before[32,39,40], the 100-times higher inoculum used for infection did not lead to a marked increase in lung viral loads at day 4 (Fig. 3C) as compared to previously used $10^3$ TCID50 standard dose (Fig. 1C, D and Fig. 2E, F). Likewise, viral RNA was readily detectable until day 12 p.i. in the lower respiratory tract of the majority (5/6) of sham treated animals, as well as in some nasal washes (2/6), suggesting protracted viral replication and shedding prior to full recovery and convalescence (Fig. 3E, F).

Importantly, all but one (23/24) YF-S0\* vaccinated hamsters showed protection against vigorous $10^5$ TCID50 VOC Beta challenge; comprising all animals with detectable nAb prior to virus exposure. Moreover, compared to sham in both single-dose and two-dose groups, viral RNA levels were significantly decreased at day 4 p.i. with an up to 6 $\log_{10}$ reduction in lungs (Fig. 3C) and, at least in the two-dose group, a 10–100-fold reduction in nasal washes (Fig. 3D). Also, with

exception of that seronegative individual, no infectious virus could be detected in the lung in any of the animals. In addition, in both vaccine groups a 10–100-fold reduction of infectious virus was observed in nasal washes; further demonstrating the high VE of YF-S0\* even under aggravated challenge conditions. Likewise, in YF-S0\* vaccinated animals the challenge virus was rapidly cleared with all detectable infectious virus eliminated by day 12 p.i., and residual viral RNA in lungs and upper respiratory tract below quantification limits (Fig. 3E, F).

When assessing VE in larger cohorts, regularly some vaccinated individuals appear to be protected against moderate challenge (Fig. 1B-D), or at least partially protected in case of more aggressive infection (Fig. 3B, E, F; day 12) despite no detectable nAb prior to challenge. Nevertheless, immunity may still have been primed. This is supported by the observation that all YF-S0\* vaccinated animals, including those few individuals which did not seroconvert prior to infection (Fig. 3B; 3/24 for prototypic spike; 1/24 or VOC Beta), showed high nAb levels already shortly after challenge infection (day 4 p.i.), whereas seroconversion in the sham group, thus without priming nor previous antigen exposure, was clearly delayed (Fig. S2).

Overall, these data corroborate the high potency of YF-S0\* to protect against vigorous VOC Beta infection. To cover most aggressive challenge employing most difficult variants, VE can be enhanced by homologous booster vaccination.

### Increased potency of vaccine candidate S0\* against Omicron variant

Immunogenicity of S0 and S0\* was tested in parallel against VOC Omicron. Whereas only in 4% (2/44) of animals vaccinated with the

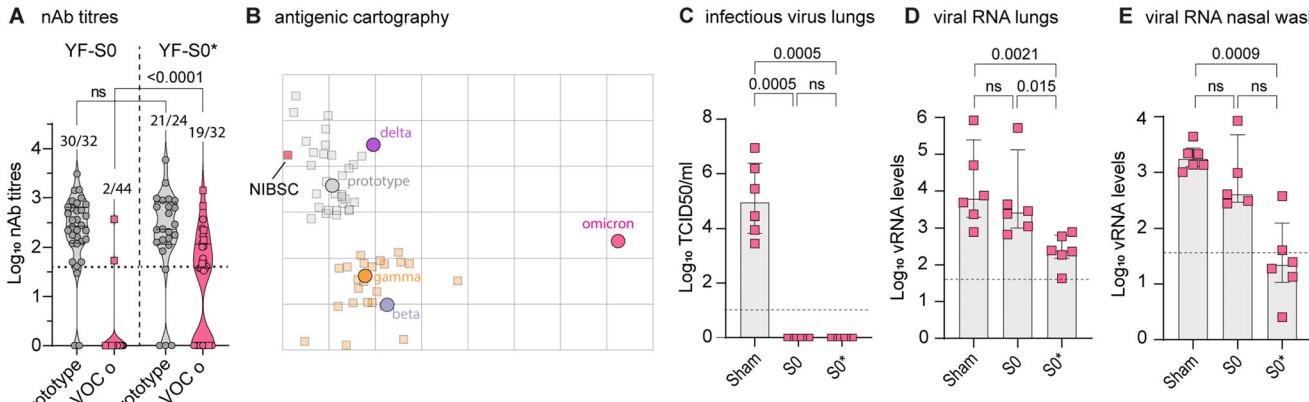

**Fig. 4 | Increased potency of vaccine candidate S0* against VOC Omicron. A** nAb titres against prototypic and omicron spike-pseudotyped virus on day 28 after vaccination with two doses (day 0 and day 7) of prototype YF-S0 or YF-S0*. NAb titres against prototypic spike-pseudotyped virus (gray circles) are from Fig. 2A and D. Same sera were retested for Omicron spike-pseudotype virus (pink symbols). Red squares denote nAb titers for those animals included in the subsequent challenge study (see also Fig. S1). **B** Antigenic cartography. Cross-reactivity of sera raised by original S0 (gray squares) and updated S0* (orange squares) vaccine antigen against five different SARS-COV-2 variants (circles: prototype, gray; VOC Beta, blue; Gamma, orange; Delta, purple; Omicron, pink) plotted on a two-dimensional map[41]. NIBSC, human reference sample (NIBSC 20/130; red square). **C–E**, Viral loads in hamster lungs (**C**, **D**) and nasal wash (**E**) 4 days after infection with $10^5$ TCID50 of VOC Omicron quantified by virus titration (**C**) and quantitative RT-PCR (**D**, **E**) (pink squares). VOC Omicron challenge was performed 7 weeks after vaccination, i.e., delayed by another 4 weeks compared to the previous standard schedule, using each $n = 6$ hamster vaccinated with YF-S0, YF-S0* or sham. Bar graphs denote median ± IQR. Dashed line represents the lower limit of quantification. Differences between groups were analyzed using nonparametric Kruskal–Wallis test uncorrected for ties.

prototypic S0 construct cross-reactive nAbs were detectable that could also neutralized the Omicron variant, vaccination with updated S0* resulted in 60% (19/32) seroconversion to such nAbs. Quantitatively, S0* resulted in a marked ~15-fold increase ($p < 0.0001$) in nAbs with activity against Omicron to $\log_{10}$ GMT of 1.2 (IQR 0–2.1); compared to S0 which resulted in hardly any nAb against Omicron ($\log_{10}$ GMT of 0.09 $\log_{10}$ IQR 0–0.09), confirming the substantial gain in immunogenicity for S0* (Fig. 4A). In addition, nAb levels remained stable over a period of 2 months (Fig. S3A, B).

This change in cross-reactivity of sera raised by the original (YF-S0) and updated (YF-S0*) vaccine antigen against five different virus variants (prototype; VOCs Beta, Gamma, Delta and Omicron) was further studied using antigenic cartography[41] (Fig. 4B). For simplicity, VOC Alpha was not considered since it did not differ from prototype virus, neither regarding VE of S0 nor nAb titre as correlate of protection (Fig. 1). Specifically, we constructed a two-dimensional projection that geometrically maps median serum neutralization titres ($SNT_{50}$) between sera and respective antigens as antigenic distances. This analysis revealed a pattern of antigenic diversification between prototype virus on the one hand and VOCs Beta and Gamma on the other hand, with VOC Delta being mapped closer to the prototype virus as compared to Beta and Gamma. This observation is consistent with recently described patterns of convergent evolution in spike for VOCs Beta and Gamma, and Delta climbing a different fitness peak[42]. VOC Omicron appeared by far more distant from any other strain, in line with recent larger scale antigenic analysis[43].

Corresponding with this visual pattern of clustering, antigenic distances for S0* sera were significantly larger to prototype and VOC Delta than to Beta and Gamma (t-test; $p < 0.001$). Intriguingly, however, this obvious antigenic drift did not reduce the overall higher potency of S0*, which included equally strong humoral responses to prototypic spike and VOC Delta (Fig. 2A, D; Fig. S1). Likewise, despite being antigenically still largely divergent from other VOC (mean range: 0.8–3.3 units), antigenic distances regarding VOC Omicron were significantly shorter (t-test; $p < 0.0001$) for S0* sera compared to S0 sera (respectively, mean ± SD; 5.5 ± 0.7 and 6.6 ± 0.5), in further support for the marked gain in Omicron-specific humoral immunity by S0* (Fig. 4A). Opposed to prototypic SARS-CoV-2 and earlier VOCs, Omicron does not cause productive infection nor apparent pathology in

hamsters after challenge with $10^3$ TCID50[44]. Using a 100-times higher inoculum of $10^5$ TCID50 resulted in substantial infectious virus titres in the lung, and readily detectable viral RNA in the lung and nasal washes of sham vaccinated animals (Fig. 4C–E). However, whereas vaccination with S0 did not result in a marked reduction in viral RNA levels in the upper or lower respiratory tract, S0*-vaccinated hamsters, showed a clear 10–100-fold reduction of viral RNA loads in their lung and nasal wash (Fig. 4D, E). Intriguingly, no infectious virus could be detected anymore in both S0 and S0* vaccinated animals (Fig. 4C); with quantitative differences between both vaccines possibly obscured by the generally poor replication of VOC Omicron in the lungs of hamsters as compared to the other virus variants[44].

Overall, these findings support the general observation that vaccines employing prototypic spike as antigen are losing serological coverage and protection, in particular towards those VOCs (Beta, Gamma and Omicron) linked to escape from antibody-mediated immunity. We further demonstrate how vaccine potency and induction of cross-reacting nAb can be markedly enhanced by alternative spike antigen choice and design.

### Blocking of VOC Delta transmission by single-dose vaccination

VOC Delta is characterized by a particular efficient human-to-human transmission[45]. An added benefit of vaccination at the population level would hence be an efficient reduction in viral shedding and transmission by vaccinated people[46], ideally from single-dose vaccination. For experimental assessment, two groups of hamsters ($n = 6$ each) were either vaccinated only once with a single dose of $10^4$ PFU of S0* or sham[28], and were intranasally infected three weeks later with a high dose comprising $10^5$ TCID50 of VOC Delta (i.e. 100× higher dose than standard before) to serve as index (donor) animals for direct contact transmission (Fig. 5A). At 2 dpi, i.e., at onset of increasing viral loads and shedding[40,47], index animals were each co-housed with one non-vaccinated sentinel for two consecutive days to maximize exposure and chance of horizontal transmission[48]. At 4 dpi, index hamsters were sacrificed, and lungs were assessed for viral RNA, infectious virus and histopathology. Sentinels were sacrificed another 2 days later and analyzed accordingly.

As expected from previous experiments, viral loads in S0*-vaccinated index animals were much lower than in non-vaccinated index

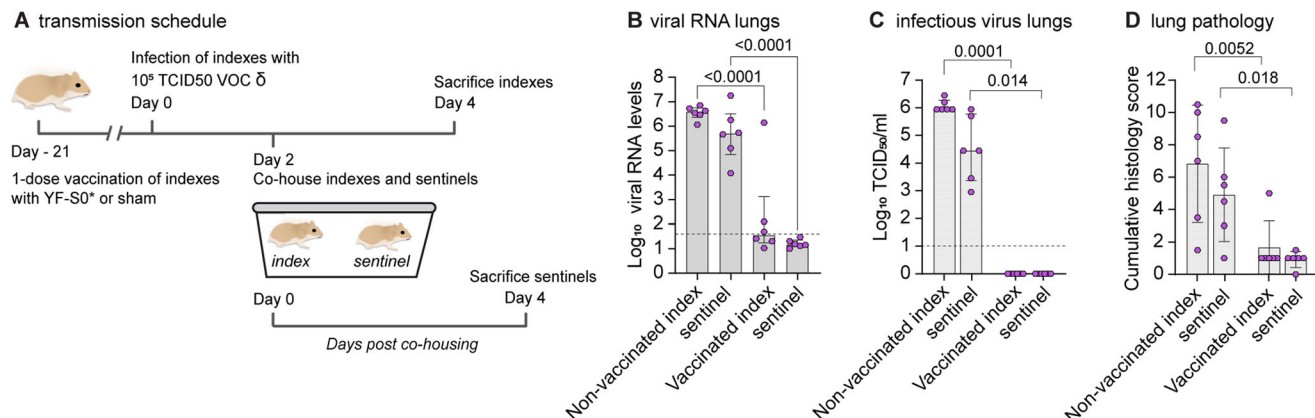

**Fig. 5 | A vaccine based on updated spike antigen S\* prevents transmission of VOC Delta.** Effect of YF-S0\* vaccination on viral transmission to non-vaccinated contact hamsters (n = 6 per group). Index hamsters were either sham vaccinated (n = 6) or vaccinated (n = 6) with a single dose of $10^4$ PFU of YF-S0\* and infected intranasally on day 21 with $10^5$ TCID$_{50}$ of VOC Delta. Two days after infection, index animals were paired and co-housed with each one naïve sentinel for 2 days. Index and sentinel animals were sacrificed each 4 days after infection or start of exposure, respectively. **A** Schematic of transmission experiment. **B**, **C**, Viral loads in hamster lungs 4 days after infection quantified by quantitative RT-qPCR (**B**) and virus titration (**C**). **D** Cumulative lung pathology scores from H&E-stained slides of lungs for signs of damage. Data for sentinels (purple circles; right) are each shown direct adjacent to their respective donor groups (left). Bar graphs denote median ± IQR. Dashed line represents the lower limit of quantification. Differences between groups were analyzed using nonparametric Kruskal–Wallis test uncorrected for ties.

animals, or than in sentinels that had been in close contact with non-vaccinated donors (Fig. 5B, C). Importantly, only very low levels of viral RNA and no infectious virus was observed in non-vaccinated sentinels that had been co-housed with S0\*-vaccinated donors. Also, lung pathology was reduced significantly in vaccinated index and co-housed sentinels as compared to sham vaccinated index and respective co-housed sentinels (Fig. 5D). To our knowledge, this constitutes the first preclinical experimental evidence for protection from SARS-CoV-2 transmission by any vaccine in the most stringent hamster co-housing model, suggesting that the block against vertical transmission conferred by S0\* could be more complete than that observed for current vaccines[49]. Nevertheless, considering the short interval between immunization and exposure, this promising result may need to be corroborated in experiments with extended schedules. The transmission protocol established here may serve as a general blueprint for future studies.

## Discussion

Little experimental support exists on how well current first-generation vaccines protect against the full spectrum of VOCs. While likely protecting from severe COVID-19 caused by any SARS-CoV-2 strain, a clear drop in VE was already observed during early clinical trials conducted in regions with high circulation of VOC Beta as paradigm of an E484K Spike variant and others known to escape nAb recognition[50]. Experimentally, such a drop in protective immunity is confirmed by higher viral loads in macaques vaccinated with an Adenovirus-vectored prototype spike antigen (Ad26.COV2.S) and challenged with VOC Beta[51]. Likewise, in the more stringent hamster model, immunity acquired during previous SARS-CoV-2 (prototype) infection, or by Ad26.COV2.S vaccination, led only to partial suppression of heterologous VOC Beta replication[52]. In latter case, replicative viral RNA was still detectable two weeks after challenge (<2 log$_{10}$ reduction compared to sham), which is in line with the observed failure of our prototypic YF-S0 to confer full protection against VOC Beta (Fig. 1C, D). By contrast, 4 days p.i. at peak of infection in sham controls, viral replication was reduced to undetectable levels for all five VOCs by YF-S0\* vaccination using an updated spike antigen, irrespective of inoculum virus ($10^3$ or $10^5$ TCID50) (Fig. 2F, Fig. 3C and Fig. 5C, index animals). Further, in YF-S0\* vaccinated animals, virus elimination appeared complete by day 12 p.i.; with no evidence for any protracted VOC Beta infection anymore neither in the upper nor lower respiratory tract, in contrast to what has been reported for Ad26.COV2.S employing prototypic spike antigen in

hamsters[52]. Finally, YF-S0\* blocked transmission of VOC Delta, for which a single-dose vaccination was sufficient (Fig. 5).

Both (i) the high incidence and hit rate of SARS-CoV-2 and (ii) increasing global vaccination coverage result in a high prevalence of antibodies in the human population with specificity for prototypic spike or spike sequences from early variants. In general, though under debate how relevant for COVID-19[53–55] antigen exposure results in an immunological imprint that upon re-vaccination may skew anamnestic responses towards previously encountered prototypic spike sequences (original antigenic sin)[56]. It is hence not clear whether or not novel variant-proof vaccines will suffer from relatively poor variant-specific nAb booster responses; a major unanswered question to be finally solved by clinical inspection. Nonetheless, following the example of influenza as paradigm of a rapidly evolving vaccine target[57], nobody would seriously consider use of historic antigen structures (e.g., from the 1918 pandemic) for current flu vaccines. Likewise, it is highly unlikely that future COVID-19 vaccines can continue to rely on early 2019/2020 sequences.

The stringent hamster model is generally well suited to assess both aspects of preclinical VE, individual protection and transmission[28,40,47,48]. A possible shortcoming of our current study is the limited infectivity of VOC Omicron in hamsters[44], and that VE of YF-S0\* against Omicron can hence not directly be assessed in this golden standard model. However, considering (i) that VOC Omicron is poorly, if at all, covered by current vaccines[58] and (ii) nAbs are strongly correlated with VE, the gain in Omicron-specific nAb achieved by YF-S0\* vaccination is remarkable. Successful testing in complementary animal models (e.g. human ACE2-transgenic hamsters)[59] will warrant further development of YF-S0\* as a second-generation COVID-19 vaccine candidate with broader coverage of relevant virus strains.

In more general terms, our findings strongly suggest that first-generation COVID-19 vaccines need to be adapted to keep up with the evolution of variants driving the ongoing global SARS-CoV-2 pandemic, employing modified spike variants that by proper antigen choice and/or advanced antigen design cover critical combinations of mutations driving both nAb escape and enhanced transmission.

## Methods
### Viruses and animals
All virus-related work was conducted in the high-containment BSL3 facilities of the KU Leuven Rega Institute (3CAPS) under licenses AMV 30112018 SBB 219 2018 0892 and AMV 23102017 SBB 219 2017 0589

according to institutional guidelines. All SARS-CoV-2 strains used throughout this study were isolated in house (University Hospital Gasthuisberg, Leuven) and characterized by direct sequencing using a MinION as described before[39]. Strains representing prototypic SARS-CoV-2 (Wuhan; EPI_ISL_407976)[39], VOCs Alpha (B.1.117; EPI_ISL_791333), and Beta (B.1.351; EPI_ISL_896474) have been described[32]. Strains representing VOCs Gamma (P.1; EPI_ISL_1091366) and Delta (B.1.617.2; EPI_ISL_2425097) were local Belgian isolates from March and April 2021, respectively. The strain representing VOC Omicron (B.1.1.529) variant was isolated from a nasopharyngeal swab taken from a traveler returning to Belgium at the end of November 2021 (B.1.1529; EPI_ISL_6794907)[60] and its initial biological characterization has been described elsewhere[44]. All virus stocks were grown on Vero E6 cells and used for experimental infections at low in vitro passage (P) number, P3 for prototype and P2 for all four VOCs. Absence of furin cleavage site mutations was confirmed by deep sequencing. Median tissue culture infectious doses (TCID50) were defined by titration as described[32,39] using Vero E6 cells (ATCC CRL-1586; generous gift from Peter Bredenbeek, LUMC, the Netherlands) as substrate, except for VOC Delta, for which A549-Dual hACE2-TMPRSS2 cells (Invitrogen Cat. No. a549d-cov2r) cells were used for a more pronounced virus induced cytopathic effect (CPE).

Housing and experimental infections of hamsters have been described[28,39,40] and procedures were approved by the ethical committee of KU Leuven (licenses P050/2020 and P55-2021), following institutional guidelines approved by the Federation of European Laboratory Animal Science Associations (FELASA). Throughout the study, 6–8-week-old female Syrian hamsters (*Mesocricetus auratus*, strain RjHan:AURA) were sourced from Janvier Laboratories and kept per two in individually ventilated isolator cages. Animals were anesthetized with ketamine/xylazine/atropine and intranasally infected with 50 μL of virus stock (25 μL in each nostril) containing either $10^3$ or $10^5$ TCID50 as specified in the text and euthanized 4 days post infection (dpi) for sampling of the lungs and further analysis if not otherwise stated (see below and Fig. 3). Animals were monitored daily for signs of disease (lethargy, heavy breathing, or ruffled fur). In contrast to others[35,47,52], in our hamster model using relatively young (8 weeks at immunization; 11 weeks at challenge) female outbred hamsters, SARS-COV-2 infection does not lead to a marked weight loss; despite it is associated with vigorous virus replication and induction of pronounced lung pathology (including cytokine elevation, pathohistological changes, radiologically confirmed consolidations)[28,32,39,40].

## Vaccine candidate

The general methodology for the design and construction of a first YF17D-based SARS-CoV-2 vaccine candidate (YF-S0) has been described[28]. Several mutations were introduced into original YF-S0 to generate second-generation vaccine candidate YF-S0*. The first series of mutations is based on the spike sequence of VOC Gamma[33]: L18F, T20N, P26S, D138Y, R190S, K417T, E484K, N501Y, D614G, H655Y, T1027I, V1176F. A second series of mutations is based on a locked spike variant, stabilizing the protein in a more immunogenic prefusion confirmation: A892P, A942P, V987P[34].

## Production of spike-pseudotyped virus and serum neutralization test (SNT)

Virus-neutralizing antibodies (nAb) were determined using a set of VSV spike-pseudotype viruses essentially as described[28]. For this purpose, five different pseudotypes were generated using expression plasmids of respective spike variants: for prototype B.1/D614G as before[28] or sourced from Invivogen for VOCs Beta (Cat. No. plv-spike-v3), Gamma (Cat. No. plv-spike-v5) and Delta (Cat. No. plv-spike-v8). The VOC Omicron spike expression construct was assembled from six custom synthetized gDNA fragments (IDT, Leuven) and cloned into pCAGGS plasmid backbone as before[28]. Briefly, depending on the plasmid

background, BHK-21J cells (generous gift from Peter Bredenbeek, LUMC, the Netherlands)[28] for variant B.1/D614G and Omicron, or HEK-293T cells (ECACC Cat. No. 12022001) for Beta, Gamma and Delta were transfected with the respective SARS-CoV-2 spike protein expression plasmids, and 1 day later infected with GFP-encoding VSVΔG backbone virus[61]. Two hours later, the medium was replaced by medium containing anti-VSV-G antibody (I1-hybridoma, ATCC CRL-2700) to neutralize residual VSV-G input. After 26 h incubation at 32 °C, the supernatants were harvested. To quantify nAb, serial dilutions of serum samples were incubated for 1 h at 37 °C with an equal volume of S-pseudotyped VSV particles and inoculated on Vero E6 cells for 19 h.

The resulting number of GFP expressing cells was quantified on a Cell Insight CX5/7 High Content Screening platform (Thermo Fischer Scientific) with Thermo Fisher Scientific HCS Studio (v.6.6.0) software. Median serum neutralization titres (SNT50) were determined by curve fitting in Graphpad Prism after normalization to virus (100%) and cell controls (0%) (inhibitor vs. response, variable slope, four parameters model with top and bottom constraints of 100 and 0%, respectively). The human reference sample NIBSC 20/130 was obtained from the National Institute for Biological Standards and Control, UK.

## Antigenic cartography

We used the antigenic cartography approach developed for influenza hemagglutination inhibition assay data to study the antigenic characteristics of the SARS-CoV-2 spikes[41]. This approach transforms SNT50 data to a matrix of immunological distances. Immunological distance $d_{ij}$ is defined as $d_{ij} = s_j - H_{ij}$, where $H_{ij}$ is the $\log_2$ titre of virus $i$ against serum $j$ and $s_j$ is the maximum observed titre to the antiserum from any antigen ($s_j = \max(H_{1j}, ..., H_{nj})$). Subsequently, a multidimensional scaling algorithm was used to position points representing antisera and antigens in a two-dimensional space such that their distances best fit their respective immunological distances. Even though distances are measured between sera raised by vaccination using specific spike antigens (and the NIBSC serum) and antigens, such an antigenic map also provides estimates of antigenic distances between the antigens themselves.

## Vaccination and challenge

**Standard setting.** COVID-19 vaccine candidate YF-S0[28] was used to vaccinate hamsters at day 0 and day 7 ($n = 32$) with a dose of $10^4$ PFU via the intraperitoneal route and control animals ($n = 18$) were dosed with MEM (Modified Earl's Minimal) medium containing 2% bovine serum as sham controls. Such a rapid two-dose immunization schedule has been established by us before for the prototypic YF-S0[28,62] to result in a more consistent immunization than single-dose vaccination, even though full development of B cell responses prior to the booster is not expected due to the short interval between doses. Despite probably suboptimal, two doses of YF-S0 given 7 days apart induced high levels of nAb and strong antiviral cellular immune responses, and resulted in protection from SARS-CoV-2 infection in cynomolgus macaques and COVID-19-like lung pathology in hamsters; in the case of macaques when using the clinically relevant subcutaneous route[28]. If not otherwise stated in the text, this two-dose schedule was pursued as benchmark, with as main scope to compare two antigens (S0 and S0*) head-to-head; following a fixed treatment regimen and standardized analytical pipeline.

Blood was drawn via the jugular vein under isoflurane anesthesia at day 21 for serological analysis and infection was done on the same day with prototype ($n = 10$ vaccinated; $n = 6$ sham), VOC Alpha ($n = 10$ vaccinated; and $n = 6$ sham), and VOC Beta ($n = 12$ vaccinated; $n = 6$ sham) with the inoculum of $10^3$ TCID50 intranasally. Protective nAb levels were calculated using logistic regression analysis in GraphPad Prism (version 9) as described[35].

Similarly, hamsters were vaccinated twice with $10^4$ PFU YF-S0* ($n = 24$) or sham ($n = 16$) at day 0 and day 7. Blood was collected at day

21 to analyze nAbs in serum, and animals were infected on day 24 with different variants, including VOCs Alpha, Beta, Gamma and Delta with the inoculum of $10^3$ TCID50 intranasally ($n = 6$ vaccinated and $n = 4$ sham vaccinated infected against each variant). Lungs were collected for analysis of viral RNA, infectious virus and for histopathological examination as described in[28]. Resulting vaccine efficacy (VE) (Fig. 1E) was calculated as [1 − (number of vaccinated animals with detectable virus) / (number of all infected animals)] × 100% per group of hamsters infected with the same virus strain, whereby a lung viral load >$10^2$ TCID50/ml was set as cutoff for infection[35].

**Vaccine challenge using high-dose VOC Beta inocula.** To compare the effect of single-dose (S0*) vaccination to the previously used short-interval two-dose regimen against the most difficult to cover VOC Beta, hamsters were immunized with either a single dose of $10^4$ PFU in one group ($n = 12$), or following the standard two-dose regimen in another group ($n = 12$) along with sham vaccinated animals ($n = 12$). Blood was collected at day 21 to analyze nAbs against D614G prototype and VOC Beta. At day 21 hamsters were infected with a dose of $10^5$ TCID50 of VOC Beta, i.e., with a 100-fold higher inoculum than before (setup 1). At day 4 post infection $N = 6$ animals were sacrificed from each group, and nasal washes and lungs were harvested to analyze viral RNA and viral load. Rest of the animals ($n = 6$, from each group) were kept longer to assess the long-term effect of infection[52] and were sacrificed at day 12 post infection. Lungs and nasal wash samples were collected for further analysis. Serum samples collected at day 4 and day 12 post infection were analyzed for nAbs.

**VOC Omicron challenge.** To assess VE of S0* against Omicron, hamsters were vaccinated twice at day 0 and day 7 with $10^4$ PFU YF-S0* ($n = 12$) and YF-S0 ($n = 12$). Blood was collected on days 28 and 48 to analyze nAbs against Omicron. In parallel, samples from previous experiments (day 21) were also tested for nAbs against Omicron. In contrast to previous challenge experiments (setup 1 and 2), hamsters were infected at a markedly later time point, i.e., on day 52 (7 weeks after immunization) with an inoculum of $10^5$ TCID50 of VOC Omicron (100-times higher than used standardly; setup 1). Lungs and nasal washes were collected for analysis of viral RNA and infectious virus at 4 days post infection.

**Viral load and viral RNA quantification**
Virus loads were determined by titration and RT-qPCR from lung homogenates was performed exactly as previously described in detail[28,39,40]. For determination of viral loads in the upper respiratory tract, virus and viral RNA was recovered from nasal washes, whereby 100 μl of PBS were flushed through each nostril using a pipette, and subsequently used for RNA extraction and viral load determination as previously described[28,39,44]. Lower limit of quantification (LLOQ) for RT-qPCR was calculated as before[28,39].

**Histopathology**
For histological examination, the lungs were fixed overnight in 4% formaldehyde, embedded in paraffin and tissue sections (5 μm) after staining with H&E scored blindly for lung damage (cumulative score of 1−3 each for congestion, intra-alveolar hemorrhage, apoptotic bodies in bronchial epithelium, necrotizing bronchiolitis, perivascular edema, bronchopneumonia, perivascular inflammation, peribronchial inflammation, and vasculitis) as previously established[32,39].

**Blocking of viral transmission**
Hamsters ($n = 6$) were vaccinated with a single dose of $10^4$ PFU (vaccinated only once), were bled at day 21 and infected with VOC Delta with $10^5$ TCID50 (100× higher than in standard setup 1), intranasally. Another group of non-vaccinated hamsters ($n = 6$) were also infected. Two days post infection index animals were co-housed with sentinels

for two days and separated after 2 days of exposure. These co-housing conditions had been established to maximize the chance of transmission in a series of pilot experiments, consistent to what others described[48]. All index animals were euthanized on day four post infection, and sentinels were sacrificed 4 days after initial exposure. Territorial fighting or aggression of animals was not observed, possibly favored by the choice of females and cage enrichment. Lungs were analyzed for viral RNA and infectious virus and subjected to histopathology.

**Statistical analysis**
All statistical analyses were performed using GraphPad Prism 9 software (GraphPad, San Diego, CA, USA). Results are presented as GM ± IQR or medians ± IQR as indicated. Data were analyzed using uncorrected Kruskal−Wallis test and considered statistically significant at $p$-values ≤0.05.

**Reporting summary**
Further information on research design is available in the Nature Research Reporting Summary linked to this article.

## Data availability
All data supporting the findings in this study are also available from the corresponding author upon request. Source data are provided with this paper.

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

## Acknowledgements

We are grateful to Prof. Michael A. Whitt (University of Tennessee Health Science Center) for generously sharing of plasmids for rescue of VSVΔG, and Dr. Maya Imbrechts and Dr. Nick Geukens (PharmAbs, KU Leuven) for help with hybridoma culture. We thank the staff of the Rega animalium for strong support. We thank Stijn Hendrickx, Jasper Rymenants, Tina van Buyten, Dagmar Buyst, Thibault Francken, and Niels Cremers for steady and timely help with analyzing of tissue samples and virus titrations. We finally thank Jasmine Paulissen, Catherina Coun, Céline Sablon, Jolien Timmermans, and Nathalie Thys (TPVC) for diligent serology assessment, vaccine stock generation, and skilled generation of plasmid constructs. Current work was supported by the Flemish Research Foundation (FWO) emergency COVID-19 fund (G0G4820N) and the FWO Excellence of Science (EOS) program (No. 30981113; VirEOS project and No. 40007527; VirEOS2), the European Union's Horizon 2020 research and innovation program (No. 101003627; SCORE project and No. 733176; RABYD-VAX consortium), the Bill and Melinda Gates Foundation (INV-00636), KU Leuven Internal Funds (C24/17/061) and the KU Leuven/UZ Leuven Covid-19 Fund (COVAX-PREC project), and European Health Emergency Preparedness and Response Authority (HERA). G.B. acknowledges support from the KU Leuven Internal Funds (Grant No. C14/18/094) and the Research Foundation—Flanders ("Fonds voor Wetenschappelijk Onderzoek—Vlaanderen," G0E1420N, G098321N). P.L. acknowledges funding from the European Research Council under the European Union's Horizon 2020 research and innovation program (grant agreement no. 725422-ReservoirDOCS) and from the EU grant 874850 MOOD. K.D. acknowledges grant support from KU Leuven Internal Funds (C3/19/057 Lab of Excellence).

## Author contributions

S.S. and K.D.: conceptualization; S.S., C.D.K., and L.B.: animal experimentation; S.S., T.V., W.K., M.R., and H.J.T.: data generation, analysis, and curation; S.S., H.J.T., and K.D.: original manuscript draft; S.S. and H.J.T.: visualization; T.V. and L.S.F.: construct design; T.V., W.K., and D.V.L.: serological analysis; R.A. and C.S.F.: VoC hamster models; B.W.: histological analysis; P.L. and G.B.: antigenic cartography; L.S.F., V.L., T.V., W.K., and P.M.: vaccine stocks and virus isolation; J.N., H.J.T., and K.D.: supervision, writing, and project administration; J.N. and K.D.: funding acquisition. All authors read, edited, and approved the final version of the manuscript.

## Competing interests

The authors declare no competing interests.
