## [Peer review file · Nature Communications]

REVIEWER COMMENTS

Reviewer #1 (Remarks to the Author):

Kai Dallmeier and colleagues report on their findings from serological assays and vaccination/challenge studies with the hamster model of COVID-19 that indicate a need to update the COVID-19 vaccine to provide better Spike immunity against infection by SARS-CoV-2 Variants of Concern. For the current study, the authors focused on their yellow fever virus vectored COVID-19 vaccine. To generate the second generation YF-S0* vaccine encoding a SARS-CoV-2 Gamma-like Spike, the investigators introduced 12 amino acid changes into the Wuhan SARS-CoV-2 Spike coding sequence. The authors should clarify if these substitutions were limited to differences in antigenic sites of the Spike or included all amino acid differences. If the latter, then the authors should indicate the number of amino acid differences between the Spike proteins of YF-S0* and SARS-CoV-2 prototype and Variants of Concern. The authors should provide a rationale for the immunization regimen in the Methods section, including justification of the 1-week interval between prime and booster immunizations. This abbreviated immunization schedule would not provide enough time for B cell responses to develop prior to the booster immunization. The authors should repeat the vaccination/challenge study to provide a longer interval between the second booster immunization and challenge infection to allow for establishment of memory B cell responses. A major limitation of the current study is the over reliance on vRNA levels to detect differences between vaccines. In particular, the authors claim an increased potency of protection by immunization with the YF-S0* vaccine; however, similar level of protection with respect to infectious viral titers was observed following immunization with either YF-S0 or YF-S0* vaccine. The authors should present the data on viral RNA quantification as genome copy numbers, which are easier for the reader to cross-reference with the infectious viral titers. For Figure 1, the authors do not discuss the lack of pulmonary virus titers consistently matching the fold-changes of vRNA levels. For Figure 1, some hamsters were “protected” despite a lack of nAb titers and some hamsters were “infected” despite nAb titers. The value of the data presented in Figure 4 on the potency of YF-S0* immunization to reduce transmission would be strengthened if the authors included a comparative transmission group of hamsters immunized with YF-S0. The authors should revise the concluding statement of the Results section (Line 233). The authors cannot draw a comparison of data from hamsters vaccinated with YF-S0* and challenged 21 days post-vaccination with clinical data from humans vaccinated with COVID-19 vaccines based on different vaccine technologies and formulations. The authors should incorporate the data presented as supplemental figures into the main manuscript.

There is concern regarding reuse of data within the figures. Figure 1 should be revised to include data presented in Figure 2A and 2B to provide a comprehensive summary of the experimental data with the YF-S0 vaccine. However, the Log nAb titers for the Prototype group in Figure 2A appear identical to the data for the same group presented in Figure 1B. The Log nAb titers for the YF-S0 Prototype and YF-S0* VCo groups in Figure 3A also appear identical to the same data for the same groups in Figure 2A. The data for the YF-S0* Prototype group (21/24) in Figure 3A seems identical to the data for the same

Prototype group in Figure 2D (21/24). The authors should clarify if these data are from the same experiment or from independent experiments.

Minor comments:

Some sections of the manuscript should be revised for clarity.

Line 32 should be revised to indicate the yellow fever-vectored vaccine.

Line 36 should be revised to indicate “vaccinated hamsters did not transmit Delta...”

Line 59 should be revised to indicate increased affinity for human and mouse ACE-2.

Line 66 should be revised to remove “so” to state “seroprevalence, despite naturally...”

Line 71 should be revised to indicate “carries by far the targets...”

Line 91 should be revised to indicate “S0* no longer transmitted...”

Line 95 should be revised to improve clarity.

Line 99 should be revised to remove the endorsement statement. The authors should state that the data indicate that YF-S0* is a potent second-generation vaccine candidate that should progress to clinical trials.

Reference 27 indicated on Line 104 is misleading as it appears the data being discussed was generated by the study described in Reference 27. The line should indicate the immunizations were performed as described previously.

Reviewer #2 (Remarks to the Author):

Sharma et al. deals in this manuscript with the issue of the important and urgent need for an updated COVID-19 vaccine which will be efficacious also for the emerging SARS-CoV-2 VOCs. The evaluation of their updated vaccine, based on YF17D (S0*), was compared to the original vaccine (S0) in the hamster model. Their main findings were:

1. The S0 can control the ancestral and VOC Alpha but not Beta.
2. S0* showed improvement compared to S0 regarding immunogenicity (nAb) and significant reduction of viral burden and lung pathology for Alpha, Beta, Gamma and Delta VOCs.
3. S0* showed increased potency against VOC omicron based on nAb of the vaccinated hamsters and reduced viral burden following infection.

4. A single S0* vaccination prevented transmission of the virus to co-cage sentinel hamsters.

Although the findings are important, there are some major issues, mainly regarding the model as was used here, that must be addressed in order to re-evaluate it for publication:

- There is no information beyond the 4 d.p.i. time point.
- The challenge dose used in all the VOCs of 10^3 (except of the Omicron 10^5) is on the very low spectrum of this model (for comparison see i.e. references 28, 31 in this manuscript; Yahalom Ronen et. al. Nat Commun 11, 6402 (2020); Francis ME et. al. PLoS Pathogens 17(7) (2021)).
- One of the main effects following infection in this model addresses body weight loss following infection. There is no data in this manuscript regarding this issue.
- A minimum 10 days experiment with significantly higher challenge dose, weight data and viral loads at the target organs should be performed in order to convince the readers regarding the efficacy of S0*.
- The effect of the vaccination on the different VOCs regarding the nasal turbinates are lacking significantly.
- There is no information regarding the viral loads in the index hamsters at the time of pairing with the sentinels.

Additional points to address:

1. The authors should carefully go over all the references and check their position and numbering (i.e. line 220, line 253, Ref #17, #41).
2. Line 66. Is the "so" needed?
3. Figure 1B, please add to the legend what are the 30/32.
4. Figure 1C, the y axis " fold change..." are they vs. the RNA levels of 10^3 ? Please add here and in the other relevant legends.
4. Line 156 "... dependingstudy." Please clarify.
5. Figure 2A (y axis) and 2D (title) NAb - nAb consistency.
6. Figure 2B - Please check whether the colors of the orange-gray points of the 3-log dilution were replaced.
7. Figure 2F - The TCID50 of the Sham VOC Beta, Gamma and Delta are on the upper limit of the test or are they the accurate numbers?
8. Figure 2H - please state the magnifications used (corresponding to the scale bar shown).
9. Figure 2H - A lower magnification image can be helpful for evaluating the impact of the treatment regarding i.e. the tissue/air ratio.

10. Line 183 – "For simplicity" is doubled.
11. Line 204 - The difference between Fig 3C, E and Fig 2SC,D should be stated.
12. Line 208 - The fact that the S0 treatment fully eliminated the infectious virus in the lungs should be (at least) addressed in the Discussion section. The data regarding the infectious virus in the nasals are highly missing.
13. Figure 3A - Does the data of the prototype is the same as Figure 2A and 2D? If yes, it should be stated.
14. Figure 3 C-E, please add to the legend the challenge dose.
15. Figure 3 - Consider changing the order of C and D according to the text, (line 203).
16. Figure 3 C-E- Statistical analysis is missing between S0 and S0*.
17. Figure 3, legend (line 627) should be 130?
18. Figure 4A - Please add to the Co-house..."for 2 days".
19. Figure 4B - why, as opposed, to the other figures the y axis starts in 0? Were all the values of all the animals exactly as the challenge dose? Please clarify.
20. Figure 4 legend - Please add the number of animals used in each group.
21. Figure 4B-D - Please differ between the sentinels on the x axis (legend and/or axis title).
22. Line 248 (1F-H) - not present...
23. Lines 249-252 - duplicate.
24. Line 337 - How was the blood drawn?
25. Lines 340 and 344 TCID50 consistency
26. Line 347 - The VE calculation relates to which data in the result section?
27. Line 355 - Nasal wash (Figure 3C and S2D) missing
28. Line 366 - The co-housing for 2 days was in the same cage? the hamsters did not fight each other?
29. Line 650 - Please add "." at the end.
30. Figure S2 - Please add the challenge dose to the legend.
31. Figure S2C,D - Statistical analysis is missing between S0 and S0*.

Reviewer #3 (Remarks to the Author):

This manuscript by Sharma and colleagues reports a YF17D vector vaccine with an updated SARS-CoV-2 spike gene sequence that better addresses the mutations SARS-CoV-2 has acquired in the course of adaptation to human infection and transmission. The authors compared the updated vaccine (S0*) with the original prototype that bore the ancestral S gene sequence (S0). They showed that the updated vaccine elicited improved neutralizing antibody response against VOCs, including the omicron variant. They also showed that hamsters vaccinated with S0* were more protected against VOC infection as well as histopathological changes in the lungs than those S0. Finally, they also showed that vaccination with S0* produced immunity that prevented virus transmission amongst hamsters.

While the experiments were well done and the effect of updated S gene sequence on the breadth of immune responses against VOCs are interesting, there are several major issues with this manuscript.

1. Presently, over 65% of the world's population have received at least one dose of covid-19 vaccine. Those unvaccinated would likely have also been exposed to at least one episode of SARS-CoV-2 infection. The likelihood that this S0* vaccine would be given, upon completion of clinical development, to SARS-CoV-2 naïve individuals is likely rare. Instead, it is more likely that this vaccine, when available, would be given as a booster. The more pertinent experiments would thus be to ask if the S0* vaccination is able to expand the repertoire of immune response against the VOCs when given to animals previously inoculated with another covid-19 vaccine, preferably one that is already in use. Indeed, that appears to be the limitation with boosting with mRNA vaccine with its S gene sequenced updated to address VOCs – booster doses simply elicited anamnestic responses produced from prior vaccination. Without addressing this issue, the findings in this study have little potential for clinical translation.
2. The use of fold change to show SARS-CoV-2 RNA load reduction is not helpful. It is unclear how the authors chose the reference point for fold change calculation. Showing absolute RNA copy numbers or RNA levels relative to a suitable housekeeping gene would be more informative.
3. The notion that S0* vaccination is able to prevent transmission is not of practical value given the limited interval between vaccination and infection. We now know that immunity levels wane in humans, making them eventually susceptible to infection and transmission, even if they remain reasonably well-protected against severe disease. Claiming that data from Figure 4 shows the potential for blocking transmission is thus premature without more extensive investigations.

REVIEWER COMMENTS

Reviewer #1 (Remarks to the Author):

Kai Dallmeier and colleagues report on their findings from serological assays and vaccination/challenge studies with the hamster model of COVID-19 that indicate a need to update the COVID-19 vaccine to provide better Spike immunity against infection by SARS-CoV-2 Variants of Concern. For the current study, the authors focused on their yellow fever virus vectored COVID-19 vaccine. To generate the second generation YF-S0* vaccine encoding a SARS-CoV-2 Gamma-like Spike, the investigators introduced 12 amino acid changes into the Wuhan SARS-CoV-2 Spike coding sequence. The authors should clarify if these substitutions were limited to differences in antigenic sites of the Spike or included all amino acid differences. If the latter, then the authors should indicate the number of amino acid differences between the Spike proteins of YF-S0* and SARS-CoV-2 prototype and Variants of Concern.

The precise sequence of the S0* antigen has been described in the M&M section (lines 373-376). For extra clarification we now also added extra details on the antigen design in the results section (lines 150-157):

In an attempt to generate a more universal variant-proof SARS-CoV-2 vaccine (YF-S0, S0*), we adapted the spike sequence in our original YF-S0 to include all 12 amino acid changes originally found in VOC Gamma (not a priori limited to known immunogenic sites)³⁰, plus three extra proline residues (A892P, A942P and V987P) to stabilize spike in a conformation favorable for immunogenicity^{31, 40}. Thus, in combination with three amino acid changes (RRAR>AAAA) ablating S1/S2 cleavage 28, in total 18/1260 amino acid residues (1.4% of prefusion spike, disregarding N-terminal signal peptide) were changed in the new antigen (S0*) compared to the prototypic spike sequence.*

New REF 30 describing the original Japanese VOC Gamma isolate (Fujino et al. EID 2021) has been introduced in both sections for full traceability.

The authors should provide a rationale for the immunization regimen in the Methods section, including justification of the 1-week interval between prime and booster immunizations. This abbreviated immunization schedule would not provide enough time for B cell responses to develop prior to the booster immunization. The authors should repeat the vaccination/challenge study to provide a longer interval between the second booster immunization and challenge infection to allow for establishment of memory B cell responses.

In response to the Reviewers comments we (1) justify the used immunization schedule and (2) address the role of booster and time interval between immunization and challenge infection as follows. Obviously, for ethical, capacity and time constraints, we could not redo as requested all previous experiments comprising originally already almost 150 animals and a comprehensive individual multiparameter downstream analysis.

1. Short interval in two-dose regimen.

We agree with the reviewer that a 1-week interval does not fit classical prime-boost immunization regimens, even not in small rodent models. In this study we continued to use this rapid likely

suboptimal immunization scheme that we established and characterized before for our prototype YF-S0 vaccine (Sanchez-Felipe et al. Nature 2021, REF 28). This is now explicitly mentioned already at the begin of the Results section (lines 106-107) and justified in greater depth in the M&M section as requested (lines 417-423) as follows:

Such a rapid two-dose immunization schedule has been established by us before for the prototypic YF-S0 (REF 28 and new REF 62) to result in a more consistent immunization than single-dose vaccination, even though due to the short interval between doses full development of B cell responses prior to the booster can not be expected. Despite thus probably suboptimal, two doses of YF-S0 given 7 days apart induced high levels of nAb and strong antiviral cellular immune responses, and resulted in full protection from SARS-CoV-2 infection in cynomolgus macaques and COVID-19-like lung pathology in hamsters; in the case of macaques when using the clinically relevant subcutaneous route (REF 27).

Accordingly, also the term “booster vaccination” has been substituted by “2-dose vaccination” in the respective Figure legends throughout the manuscript (see e.g., Fig. 1A).

Furthermore, the main purpose of the study was hence not to introduce or propose any new immunization schedule, as explored e.g. for rabies immunization such as the rapid and dose-sparing Thai Red Cross (Khawplod et al. Vaccine. 2006 PMID: 16494972) or Belgian Army (Soentjens & Croughe M. J Travel Med. 2021 PMID: 33009803) schedules. By contrast, as main scope of the study we compared two antigens S0 and S0* head-to-head following a fixed and standardized treatment as now explicitly mentioned (lines 423-425):

If not otherwise stated in the text, this two-dose schedule was pursued as benchmark to compare two antigens (S0 and S0) head-to-head, with as main scope to compare two antigens (S0 and S0*) head-to-head; following a fixed treatment regimen and standardized analytical pipeline.*

2. Single-dose, interval and duration.

The issue of single-dose immunization and duration of immunity has been addressed experimentally by us (i) in the original submission in the study of VOC Delta transmission study (Fig. 5), (ii) VOC Omicron VE study (Fig. 4), as well as (iii) the new added study arm assessing VE of YF-S0* against VOC Beta (completely new data set presented in new Fig. 3).

In the VOC Delta transmission study (Fig. 5), animals were vaccinated using only a single dose with as result full protection from transmission from vaccinated animals to naïve sentinels. These aspects have been now explicitly mentioned repeatedly in both the Results section (lines 275-280) as well as in a more structured M&M section (lines 472-481). Notably one dose was sufficient to protect from high 10^5 TCID50 challenge by VOC Delta (Fig. 5) and Beta (new Fig. 3). Likewise, and in contrast to the standard vaccination schedules (i.e. challenge 3 weeks after vaccination), VOC Omicron challenge was performed seven weeks (approximately 2 months) after vaccination. This information is now explicitly mentioned in the figure legend (lines 793-794), and may have escaped the reader’s attention in the previous version of the manuscript.

These previously presented data have been supplemented with a newly added set of experiments (new Fig. 3 and lines 187-231 and M&M lines 439-449) in which we further corroborate the high degree of protection conferred by our “variant-proof” antigen against the most difficult to cover VOC Beta variant

using (i) a high 10^5 TCID₅₀ challenge dose and (ii) exploring single-dose immunization schedule. We show that even under these stringent conditions full protection may be possible, demonstrated by absence of infectious virus in the lungs and nasal washes at day 4 post challenge. As expected, we also could no longer detect virus at day 12, confirming rapid virus clearance. With this regard, our new antigen seems to clearly outcompete previous attempts published by Tostanoski et al. STM 2021 (cited as REF #52), at least when critically looking on the somewhat limited data (overall wellbeing by weight, histology and qPCR after convalescence at day 14d) they disclose to claim protection against VOC Beta by the Ad26.COVS vaccine and as far as studies can be compared between labs.

A major limitation of the current study is the over reliance on vRNA levels to detect differences between vaccines. In particular, the authors claim an increased potency of protection by immunization with the YF-S0* vaccine; however, similar level of protection with respect to infectious viral titers was observed following immunization with either YF-S0 or YF-S0* vaccine. The authors should present the data on viral RNA quantification as genome copy numbers, which are easier for the reader to cross-reference with the infectious viral titers.

The representation of viral RNA loads has been adjusted to copy numbers throughout the manuscript as suggested. Accordingly, Fig. S2C,D (reporting these vRNA values in support of the fold-changes originally shown in the corresponding main figure) has now been deleted and incorporated into Fig. 3.

Regarding the concern on use of vRNA data we remain, however, a bit puzzled. Since the rise of COVID-19, qPCR is golden standard for diagnosis (see e.g. El Jaddaoui et al. Expert Rev Mol Diagn. 2021 PMID: 33593219) and vRNA data accepted as relevant proxy for SARS-CoV-2 virus loads; in current clinical practice as well as in animal models (see all references cited using animals model, including some relying exclusively on vRNA as only viral marker). Nevertheless, appreciating the limits of qPCR when employed as sole methodology, we expand our quantitative assessment and comparison of VEs to a multiparameter analysis, using several complementary method (titers, vRNA, nAb). We are convinced that this combination of several orthogonal methods provides an overall picture to show an improvement of S0* over S0, specifically an increased coverage towards 'difficult-to-cover' VOCs. For individual VOCs, evidence per parameters may vary linked relative to the virus strain studied.

For instance measuring vRNA by sensitive qPCR suggest an improvement against VOC Alpha, Beta, Gamma and Omicron, but not necessarily for prototype nor VOC Delta. Likewise, already nAb titers clearly suggest better catch-up with VOC Beta, Gamma and Omicron. Latter VOV Omicron is also a showcase for a situation where virus isolation needs to remain non-conclusive due to its limited infectivity in the available models (Abdelnabi et al. 2022 REF 44 and Halfmann et al. 2022 REF 59). In new Fig. 3 assessing vigorous VOC Beta infection-challenge, our analysis extends from day 4 and day 12, for both upper and lower respiratory tract, fully in line and complementing our previously presented data for VOC Beta. We hence do not see any inconsistencies, nor an over reliance on a single methodology. To illustrate and endorse our position we add here an – obviously simplifying – overview in following table (Table1):

Table 1: Comparison of S0* versus S0 (Multiparameter matrix)

Parameter	prototype	Alpha	Beta	Gamma	Delta	Omicron	Source
nAb	S0*=S0	n.d.t.	S0*>>S0	S0*>S0	S0*=S0	S0*>>S0	Fig. 2A vs. D; Fig. S1, Fig. S2
Virus isolation	S0*=S0	n.d.t.	S0*>>S0	n.d.t.	n.d.t.	?	Fig. 1D vs. Fig. 2F; Fig. 4C
Virus RNA	S0*=S0	S0*>S0	S0*>>S0	n.d.t.	n.d.t.	S0*>>S0	Fig. 1c vs. Fig. 2E; Fig. 4D+E

n.d.t – not directly determined

? – overall virus load even in sham animals 2Log10 lower than in other VOC; reduction readily below LOD

Moreover, with the new presentation of absolute vRNA levels, in general the correlation between vRNA and infectious titers becomes visually more consistent. However, as well established, including in the human model, vRNA levels do not necessarily match infectious virus titers recovered from the same samples. This is due to (i) the much higher sensitivity of qPCR as compared to virus isolation, and importantly (ii) vRNA detected – particularly long after exposure to the virus – may represent residual viral genomes from cellular debris (*'molecular scar'*). In such case vRNA serves only as proxy, but not proof for productive infection based on active virus replication. This is also likely the case in our data in new Fig. 3 (long term follow-up) and Fig. 4 (VE against VOC Omicron, previously Fig. 3). Comparing relative changes between groups may be hence more relevant than a full match in absolute changes (fold-changes). We report both parameters in parallel for full transparency when comparing prototypic YF-S0 and updated YF-S0* vaccines (e.g. Fig. 1D,E vs. Fig. 2E,F).

For Figure 1, the authors do not discuss the lack of pulmonary virus titers consistently matching the fold-changes of vRNA levels.

To address this issue in the revised manuscript, a brief discussion on the ratio between vRNA and TCID50 data as presented in Fig. 1 has been added (lines 114):

a marked reduction in viral RNA and <in particular> of infectious virus loads down to undetectable levels (up to 6 log₁₀ reduction) was observed

and (lines 122-127):

Also in latter comparison, virus RNA levels followed the same trend, though individual fold-changes in viral RNA levels (Fig. 1C, D and Fig. S1) did not always match the respectively observed reduction in virus titers. This is not unexpected considering (i) the higher sensitivity and dynamic range of qPCR, and likewise (ii) the general observation that viral RNA detected by qPCR may represent residues originating from cellular debris rather than viral genomes actively involved in an ongoing productive infection.

For Figure 1, some hamsters were “protected” despite a lack of nAb titers and some hamsters were “infected” despite nAb titers.

Despite vaccination, not all animals seroconverted to detectable nAb titres as stated. However, despite this lack of detectable nAb they had a reasonable chance to be protected against challenge (at least following the definition of >10² TCID50/ml of lung tissue established by van der Lubbe et al. 2021 REF

35). We have observed this in many experiments. In case of absence of detectable nAbs at day 21, challenge may serve as boost and quickly increase nAbs titres which eventually leads to protection from challenge by rapidly clearing the viruses from the system.

Such lack of a full match between nAb and protection is inherent to the definition of a “correlate of protection, CoP”, which is not to be mixed up with a mechanistic surrogate (see e.g. Barrett. JID 2020 doi:10.1093/infdis/jiz379). Considering nAb as CoP, a certain threshold of nAb will (mathematically) correlate with a certain change of being protected from infection.

Mechanistically this mismatch can be explained by the data presented in the new Fig. 3 and accompanying Fig. S2. Here, also those few YF-S0* vaccinated animals that were seronegative prior to challenge (Fig. 3B, Fig S2), show rapid re-call nAb responses following infection (day 4p.i.), whereas in sham animals, nAb titres were detected only at the latest timepoint tested (day 12p.i.). Hence, vaccination may lead to an efficient priming, despite the absence of nAb titres, humoral immunity that kicks in sufficiently fast after virus exposure to curb infection.

This finding is mentioned in line 219-224:

Nevertheless, immunity may still have been primed. This is supported by the observation that all YF-S0 vaccinated animals, including those few individuals which did not seroconvert prior to infection (Fig. 3B; 3/24 for prototypic spike; 1/24 or VOC Beta), showed high nAb levels already shortly after challenge infection (day 4 p.i.), whereas seroconversion in the sham group, thus without priming nor previous antigen exposure, was clearly delayed (Fig. S2).*

The value of the data presented in Figure 4 on the potency of YF-S0* immunization to reduce transmission would be strengthened if the authors included a comparative transmission group of hamsters immunized with YF-S0. The authors should revise the concluding statement of the Results section (Line 233). The authors cannot draw a comparison of data from hamsters vaccinated with YF-S0* and challenged 21 days post-vaccination with clinical data from humans vaccinated with COVID-19 vaccines based on different vaccine technologies and formulations.

We convene with the reviewer that the data presented in our transmission study are not exhaustive as they do not include a direct comparison between S0 and S0*. We however strongly believe that they are unique and provide an important conceptual and complementary extra to our core data set focuses on such a direct head-to-head comparison. In particular, our studies should suffice to demonstrate the need as well as the feasibility to develop transmission blocking second-generation vaccines. Therefore, we strongly prefer to keep these original data, and no new data regarding transmission were added. However, acknowledging the concern of the Reviewer, we temper our original statement mentioning the gap in knowledge due to the lack of comparative studies (neither from YF-S0, nor from licensed vaccines) as follows (line 288-294):

To our knowledge, this constitutes the first experimental preclinical evidence for full protection from SARS-CoV-2 transmission by any vaccine in the stringent hamster model, almost suggesting that the block conferred by S0 could be more complete than that observed by current vaccines.*

The authors should incorporate the data presented as supplemental figures into the main manuscript.

As requested, RNA copy numbers originally presented in the Supplementary Figures are now incorporated in main figures.

There is concern regarding reuse of data within the figures. Figure 1 should be revised to include data presented in Figure 2A and 2B to provide a comprehensive summary of the experimental data with the YF-S0 vaccine. However, the Log nAb titers for the Prototype group in Figure 2A appear identical to the data for the same group presented in Figure 1B. The Log nAb titers for the YF-S0 Prototype and YF-S0* VCo groups in Figure 3A also appear identical to the same data for the same groups in Figure 2A. The data for the YF-S0* Prototype group (21/24) in Figure 3A seems identical to the data for the same Prototype group in Figure 2D (21/24). The authors should clarify (hint to legend) if these data are from the same experiment or from independent experiments.

In the original version of the manuscript, for reasons of comparison, sometimes data from another figure were reused. Despite already indicated whether data was reused from another figure, we now specify the reuse of serological data more explicitly throughout the figure legends.

Minor comments:

Some sections of the manuscript should be revised for clarity.

Line 32 should be revised to indicate the yellow fever-vectored vaccine.

Line 36 should be revised to indicate “vaccinated hamsters did not transmit Delta...”

Line 59 should be revised to indicate increased affinity for human and mouse ACE-2.

Line 66 should be revised to remove “so” to state “seroprevalence, despite naturally...”

Line 71 should be revised to indicate “carries by far the targets...”

Line 91 should be revised to indicate “S0* no longer transmitted...”

Line 95 should be revised to improve clarify.

Line 99 should be revised to remove the endorsement statement. The authors should state that the data indicate that YF-S0* is a potent second-generation vaccine candidate that should progress to clinical trials.

Reference 27 indicated on Line 104 is misleading as it appears the data being discussed was generated by the study described in Reference 27. The line should indicate the immunizations were performed as described previously.

Many thanks for the careful reading and suggestions. All minor comments have been answered/implemented in the revised manuscript text.

Regarding the comment on line 95 (old numbering; new line 99-100), we hope that the Reviewer understands that for us an in-depth description of the shared “pool” and combinations of spike mutations in the evolution of SARS-CoV-2 VOCs and the role of “evolutionary trajectories” may be beyond the scope of the current study. For clarification, we provide several citations as new references REF 32,33,34, including with contribution from the group of collaborators, dealing with this matter in detail. Likewise, the term '*driver mutations*' that is more commonly used in cancer research has been omitted to avoid any confusion, not to suggest any genetic linkage among strains (see also line 337).

Reviewer #2 (Remarks to the Author):

Sharma et al. deals in this manuscript with the issue of the important and urgent need for an updated COVID-19 vaccine which will be efficacious also for the emerging SARS-CoV-2 VOCs. The evaluation of their updated vaccine, based on YF17D (S0*), was compared to the original vaccine (S0) in the hamster model. Their main findings were:

1. The S0 can control the ancestral and VOC Alpha but not Beta.
2. S0* showed improvement compared to S0 regarding immunogenicity (nAb) and significant reduction of viral burden and lung pathology for Alpha, Beta, Gamma and Delta VOCs.
3. S0* showed increased potency against VOC omicron based on nAb of the vaccinated hamsters and reduced viral burden following infection.
4. A single S0* vaccination prevented transmission of the virus to co-cage sentinel hamsters.

Although the findings are important, there are some major issues, mainly regarding the model as was used here, that must be addressed in order to re-evaluate it for publication:

- There is no information beyond the 4 d.p.i. time point.

Such data have been generated for day 12 using VOC Beta as a particularly difficult to cover variant and added as new Fig. 3 and Fig. S2 and extensively described (lines 187-227 and M&M, lines 439-449) and discussed (lines 311-313), also in respect of a previous study that can be considered as benchmark (Tostanoski et al. STM 2021, REF #52).

- The challenge dose used in all the VOCs of 10^3 (except of the Omicron 10^5) is on the very low spectrum of this model (for comparison see i.e. references 28, 31 in this manuscript; Yahalom Ronen et. al. Nat Commun 11, 6402 (2020); Francis ME et. al. PLoS Pathogens 17(7) (2021)).

In new Fig. 3 we showcase that a 100-times higher VOC Beta dose (10^5 TCID₅₀) does not necessarily lead to higher peak virus levels at day 4 (cf. Fig. 1 and Fig. 2), fully in line with what we demonstrated before (Kaptein et al. PNAS 2020 REF 40; Abdelnabi et al. EBioMed 2021 REF 29). Also VOC Delta challenge in YF-S0* vaccinated index/donor animals was already done with such a high 10^5 TCID₅₀ dose (old FIG-4, new FIG-5), with good VE and importantly following immunization schedule reduced to one-shot regimen. This has now been emphasized in both main text (line 186-190 and 277), M&M (lines 444-445 and 474, respectively) besides in the respective figure legends.

- One of the main effects following infection in this model addresses body weight loss following infection. There is no data in this manuscript regarding this issue.

A lack of (significant) weight change is a recurrent observation in our hamster model. Many colleagues criticize this as shortcoming. Some studies even use weight drop (as consequence for the distress caused by productive SRAS-CoV-2 infection in hamsters) as key read out (e.g. Tostanoski et al. STM 2021; cited as REF#52). Here, we honestly disagree. Likewise, also the weight drop seen consistently in the widely used K18-ACE2 mouse is likely non-physiologic (in particular, compared to the human and non-human primate model); in former case as consequence of virus-induced encephalitis and paralysis. Instead of focusing on weight, we try to provide compelling evidence for VE by a comprehensive and

quantitative multiparameter virological (virus titer), molecular (qPCR) and histopathological assessment to support our hypothesis. For new Fig. 3 this now also includes data for day 4 and day 12, for both upper and lower respiratory tract. We do not see any inconsistencies, nor an over reliance on a single methodology. Also, in contrast to the model used by for instance the Barouch group (Tostanoski et al. Nat Med. 2020), we use consistently younger outbred animals with *ad libitum* feeding that seem to be more resistant to such a weight loss as prime infection outcome.

In the revised manuscript the weight conundrum is discussed as follows (lines 366-370):

In contrast to others ^{36, 48, 61}, *in our hamster model using relatively young (8 weeks at immunization; 11 weeks at challenge) female outbred hamsters, SARS-COV-2 infection does not lead to a marked weight loss; despite it is associated with vigorous virus replication and induction of pronounced lung pathology (including cytokine elevation, pathohistological changes, radiologically confirmed consolidations)* ^{28, 29, 39, 40}.

- A minimum 10 days experiment with significantly higher challenge dose, weight data and viral loads at the target organs should be performed in order to convince the readers regarding the efficacy of S0*.

Please see our previous answer. Data for day 4 and day 12, for both upper and lower respiratory tract were now included as well to fully assess VE of S0*.

- The effect of the vaccination on the different VOCs regarding the nasal turbinates are lacking significantly.

Besides for the data showing an improved VE of S0* over S0 against VOC Omicron in nasal washes (Fig. 4, old Fig. 3), we now included a new Fig. 3 as well, showing virological and molecular analysis of nasal washes for VOC Beta (high dose VOC Beta; 1-dose vs 2-dose YF-S0*, 4 day and 12 day p.i.). A repeat to demonstrate VE for all VOC regarding nasal turbinates was obviously not feasible for ethical, capacity and time constraints. However, our old data comprise samples from originally already almost 150 animals and a comprehensive individual multiparameter downstream analysis that is fully consistent with these new complementary data presented in Fig. 3.

- There is no information regarding the viral loads in the index hamsters at the time of pairing with the sentinels.

Such data are not available. However, we (e.g. Kaptein et al. PNAS 2020, REF #40) and others established critical parameters for efficient transmission (timepoint and duration of co-housing). In general, direct contact transmission in the hamster model is highly efficient. In our case 100%, also considering the high viral inocula used for challenge of the donor/index animals (10^5 TCID₅₀) and the high resulting viral loads (Fig. 5B,C) and pathology (Fig. 5D) in non-vaccinated donors at day 4 p.i. A rationale is now included in the M&M section (line 479-480) as follows:

These co-housing conditions had been established to maximize the chance of transmission in a series of pilot experiments, consistent to what others described ⁴⁹.

Additional points to address:

- 1. The authors should carefully go over all the references and check their position and numbering (i.e. line 220, line 253, Ref #17, #41).** We carefully checked all references and their position and numbering
- 2. Line 66. Is the "so" needed?** The word "so" has been removed
- 3. Figure 1B, please add to the legend what are the 30/32.** Information has been added to the legend
- 4. Figure 1C, the y axis " fold change..." are they vs. the RNA levels of 10³? Please add here and in the other relevant legends.** Data are now presented as Log₁₀ vRNA levels.
- 4. Line 156 "... dependingstudy."** For clarity, this sentence has been rephrased
- 5. Figure 2A (y axis) and 2D (title) NAb - nAb consistency.** Checked and adapted where necessary
- 6. Figure 2B - Please check whether the colors of the orange-gray points of the 3-log dilution were replaced.** Indeed, the colors have been fixed.
- 7. Figure 2F - The TCID₅₀ of the Sham VOC Beta, Gamma and Delta are on the upper limit of the test or are they the accurate numbers?** These are accurate numbers as reconfirmed by careful re-analysis of the original titration data.
- 8. Figure 2H - please state the magnifications used (corresponding to the scale bar shown).** Information has been added in the legend.
- 9. Figure 2H - A lower magnification image can be helpful for evaluating the impact of the treatment regarding i.e. the tissue/air ratio.** These data were not available, may however be considered supplementary to the already presented scoring.
- 10. Line 183 – "For simplicity" is doubled.** Doubling has been removed
- 11. Line 204 - The difference between Fig 3C, E and Fig 2SC,D should be stated.** New figures have been added. Figure 3 is now Figure 4. Figure S2 is now Figure S3. The effect of vaccination as fold change has been replaced by showing the total vRNA levels.
- 12. Line 208 - The fact that the S0 treatment fully eliminated the infectious virus in the lungs should be (at least) addressed in the Discussion section.** We added some discussion to line 263-265 as suggested :
Intriguingly, no infectious virus could be detected anymore in both S0 and S0 vaccinated animals (Fig. 4C); <with quantitative differences between both vaccines possibly obscured by the generally poor replication of VOC Omicron in the lungs of hamsters as compared to the other virus variants (REF 45).>*
- 13. Figure 3A - Does the data of the prototype is the same as Figure 2A and 2D? If yes, it should be stated.** Yes, this is now mentioned more explicitly in the legend to the figures.
- 14. Figure 3 C-E, please add to the legend the challenge dose.** This information has now been added to the legend.
- 15. Figure 3 - Consider changing the order of C and D according to the text, (line 203).** We now changed the order of the panels.

16. **Figure 3 C-E- Statistical analysis is missing between S0 and S0***. Additional statistical analysis has been done and added to the figure.
17. **Figure 3, legend (line 627) should be 130?** This has been corrected.
18. **Figure 4A - Please add to the Co-house..."for 2 days"**. This information has been added.
19. **Figure 4B - why, as opposed, to the other figures the y axis starts in 0? Were all the values of all the animals exactly as the challenge dose? Please clarify.** In all other figures, except new Fig. 4C, vRNA levels start at 0. For conformity, we have replaced Figure 4C by a new panel in which Y axis starts at 0 as well.
20. **Figure 4 legend - Please add the number of animals used in each group.** This information has been added to the legend.
21. **Figure 4B-D - Please differ between the sentinels on the x axis (legend and/or axis title).** This information has been added to the legend.
22. **Line 248 (1F-H) - not present...** Reference to the correct figure panels has been adjusted.
23. **Lines 249-252 – duplicate.** Duplication has been removed.
24. **Line 337 - How was the blood drawn?** Information has been added to the M&M section (line 426).
25. **Lines 340 and 344 TCID50 consistency.** TCID50 has now been written consistently throughout the manuscript and figures.
26. **Line 347 - The VE calculation relates to which data in the result section?** This information has now been added.
27. **Line 355 - Nasal wash (Figure 3C and S2D) missing.** This information has been added to the M&M section, lines 459-463.
28. **Line 366 - The co-housing for 2 days was in the same cage? the hamsters did not fight each other?** Additional information has been added on the co-housing conditions and animal behavior (see line 479-480).
29. **Line 650 - Please add "." at the end.** This has been added.
30. **Figure S2 - Please add the challenge dose to the legend.** Information on the challenge dose is now included in all figures.
31. **Figure S2C,D - Statistical analysis is missing between S0 and S0***. This data is now included in Figure 4 and statistical analysis between S0 and S0* has been performed.

Reviewer #3 (Remarks to the Author):

This manuscript by Sharma and colleagues reports a YF17D vector vaccine with an updated SARS-CoV-2 spike gene sequence that better addresses the mutations SARS-CoV-2 has acquired in the course of adaptation to human infection and transmission. The authors compared the updated vaccine (S0*) with the original prototype that bore the ancestral S gene sequence (S0). They showed that the updated vaccine elicited improved neutralizing antibody response against VOCs, including the omicron variant. They also showed that hamsters vaccinated with S0* were more protected against VOC infection as well as histopathological changes in the lungs than those S0. Finally, they also showed that vaccination with S0* produced immunity that prevented virus transmission amongst hamsters.

While the experiments were well done and the effect of updated S gene sequence on the breadth of immune responses against VOCs are interesting, there are several major issues with this manuscript.

We appreciate the positive judgment regarding the quality of our experiments to demonstrate an improvement of vaccine immunogenicity (breadth of variant coverage) and efficacy (incl. against pathology and transmission). We also agree that several questions remain unsolved that would increase the potential of clinical translation for our candidate vaccine. Obviously also other vaccine platforms are facing issues of “primary antigenic sin” (point #1). Also, how we aim at demonstrating prevention of transmission here (point #3) may remain preliminary.

However, despite relevant on its own, we are truly convinced that answering these open points #1 and #3 in additional exhaustive animal experiments is out of scope of the current study. The focus of the current study is the direct head-to-head comparison of the vaccine efficacy of regarding emerging variants studying the performance of (i) an established first-generation vaccine antigen and (ii) a second-generation antigen candidate. Though not exhaustive, our complementary transmission studies should suffice to demonstrate the urgency as well as feasibility to develop transmission blocking vaccines. By this means we also hope to endorse and stipulate an important new path to be included in the preclinical testing of future vaccine candidates.

Currently approved first-generation COVID vaccines have been designed based on a 2019 antigen sequence and clinically developed as one-to-two shot vaccines to prevent symptomatic SARS-CoV-2 infection; largely based on the induction of nAbs as proposed correlate of protection. Due to the unprecedented evolution of the pandemic, their product profile has changed: currently clinical practice is considering a 3rd and 4th booster-dose (every 3-6 month) with as aim to at least prevent severe COVID, hospitalization and death; proposedly by cell-mediated immunity in the light of waning humoral immunity (Tan et al. Cell Rep. 2021; Bonifacius et al. Immunity. 2021). Thus, we consider that there is a lot of space for improvement, including establishing vigorous conceptual evidence in stringent preclinical models.

In the current study, the main message remains that there is an urgency to develop second generation COVID-19 vaccines (i.e. “COVID-22+ vaccines”) with improved specifications (variant-proof; ideally single-shot and with demonstration of impact on transmission of variants). Importantly, to the best of our knowledge, until now no team (for obvious reasons definitely none of those teams behind of any commercialized first-generation COVID vaccine) ever showed to what large extent the current prototype vaccine may truly fail against at least some variants (as clearly we do in Fig. 1 for VOC Beta)

in stringent preclinical models. One of the few studies (if not the only available on the matter) by Tostanoski *et al.* (STM 2021; cited as REF 52) demonstrates partial protection against VOC Beta by the JNJ Ad26.COVID.S vaccine. Importantly, the authors do not disclose any virological, nor molecular, nor histological data for the most aggressive acute phase of infection (around day 4 p.i.); only weight evolution and reduction (sic! not clearance) of vRNA during convalescence (14d p.i.). Both read-outs may be judged as suggestive for some level of protection against severe disease, though elegantly avoiding disclosure of quantitative multi-parameter assessment of VE and interpreted as evidence for good coverage of their prototypic vaccine against heterologous variants.

In summary, our study is unique because we show both (i) the large drop in immunogenicity as well as VE of old antigen against variants, together with (ii) a possible antigen design how to catch up with all 5 identified VOC (i.e. not only a selection).

1. Presently, over 65% of the world's population have received at least one dose of covid-19 vaccine. Those unvaccinated would likely have also been exposed to at least one episode of SARS-CoV-2 infection. The likelihood that this S0* vaccine would be given, upon completion of clinical development, to SARS-CoV-2 naïve individuals is likely rare. Instead, it is more likely that this vaccine, when available, would be given as a booster. The more pertinent experiments would thus be to ask if the S0* vaccination is able to expand the repertoire of immune response against the VOCs when given to animals previously inoculated with another covid-19 vaccine, preferably one that is already in use. Indeed, that appears to be the limitation with boosting with mRNA vaccine with its S gene sequenced updated to address VOCs – booster doses simply elicited anamnestic responses produced from prior vaccination. Without addressing this issue, the findings in this study have little potential for clinical translation.

We agree that pre-existing immunity for COVID-19 may skew any further booster response, including by any second-generation vaccine. However, it is not clear whether or not vaccination against SARS-CoV-2 suffers from “original antigen sin”. Despite evidence for imprinting, there is emerging promising evidence that antibody breadth against viral variants elicited by vaccination (sic! Using prototype antigen) is superior than that elicited by natural infection and tends to improve over several months (Röltgen *Cell*. 2022 REF #53). Overall, point #1 may hence be considered a relevant follow-up question, yet in a second step. Here we present the performance of the new antigen candidate as such.

Also, we honestly disagree that the fact that 65% of all humans has been offered already at least one COVID-19 vaccine dose would not leave space for the development of new vaccines based on novel platforms and following an advanced antigen design. In fact, current global vaccine coverage leaves 35+% percent (>2 billion people) without proper COVID-19 immunity. Moreover, second- and third-generation vaccines will be needed, considering the risk that future antiviral immune responses could be skewed by current first-generation vaccines (and the platforms employed for immunization) do not solve this emerging issue, rather make it more burning.

To address the issue of imprinting we now add following to the discussion, including newly added references (lines 316-326):

Both (i) the high incidence and hit rate of SARS-CoV-2 and (ii) increasing global vaccination coverage result in a high prevalence of antibodies in the human population with specificity for prototypic spike

or spike sequences from early variants. In general, though under debate how relevant for COVID-19 (REF #54-56) antigen exposure results in an immunological imprint that upon re-vaccination may skew anamnestic responses towards previously encountered prototypic spike sequences (original antigenic sin) (REF #57). It is hence not clear whether or not novel variant-proof vaccines will suffer from relatively poor variant-specific nAb booster responses; a major unanswered question to be finally solved by clinical inspection. Nonetheless, following the example of influenza as paradigm of a rapidly evolving vaccine target (REF #57), nobody would seriously consider use of historic antigen structures (e.g. from the 1918 pandemic) for current flu vaccines. Likewise, it is highly unlikely that future COVID-19 vaccines can continue to rely on early 2019/2020 sequences.

2. The use of fold change to show SARS-CoV-2 RNA load reduction is not helpful. It is unclear how the authors chose the reference point for fold change calculation. Showing absolute RNA copy numbers or RNA levels relative to a suitable housekeeping gene would be more informative.

We now changed to show viral RNA copy numbers, in line also with a similar request of Reviewer #1.

3. The notion that S0* vaccination is able to prevent transmission is not of practical value given the limited interval between vaccination and infection. We now know that immunity levels wane in humans, making them eventually susceptible to infection and transmission, even if they remain reasonably well-protected against severe disease. Claiming that data from Figure 4 shows the potential for blocking transmission is thus premature without more extensive investigations.

We understand the preliminary character of the transmission experiments presented here and tried to temper our claims and conclusions as cited in the following (line 288-294), also in line with similar concerns raised by Reviewer #1. As explained before a more extensive investigation in that direction can be considered out of scope for the current study, we keep but do not amend this study arm with more experimental data in the current manuscript.

To our knowledge, this constitutes the first <preclinical> experimental evidence for full protection from SARS-CoV-2 transmission by any vaccine <in the stringent hamster model, almost suggesting that the block conferred by S0 could be more complete than that observed by current vaccines⁵¹. Nevertheless, considering the short interval between immunization and exposure, this promising result may need to be corroborated in experiments with extended schedules. The transmission protocol established here may serve as a general blueprint for future studies.>*

REVIEWERS' COMMENTS

Reviewer #1 (Remarks to the Author):

The manuscript submission by Dallmeier and colleagues reports on pre-clinical assessment of their yellow fever virus-based COVID-19 vaccine in the hamster model. The authors demonstrate that 18 amino acid changes of the prototypic spike sequence expressed by their vaccine backbone broadens protective antibody responses against multiple lineages. The findings of the authors support updating of the licensed COVID-19 vaccines, and are in agreement with recently authorized clinical use of bivalent COVID-19 vaccines. A major strength of their vaccination approach is likely induction of mucosal immunity against SARS-CoV-2 infection. A major weakness of the experimental approach is the abbreviated sequential vaccination schedule with 7-day interval and challenge infection at 21 days post vaccination. This abbreviated vaccination schedule does not allow establishment of memory responses following prime immunization or recall of memory responses. In general, the manuscript requires proofreading to improve clarity in some parts of the text. Do the 2nd and 3rd paragraphs of Vaccination and challenge, (1) standard setting, provide redundant information? The authors should revise the third sentence or the Abstract to indicate their yellow fever virus vaccine platform.

Reviewer #2 (Remarks to the Author):

The revised manuscript by Sharma et al. was revised in response to the comments provided.

Most of my concerns have been addressed in the revision and the new added data shown in Figure 3 strengthen the overall message that the YF-S0* suppressed YF-S0 vaccine and that the first generation vaccines should be updated against emerging VOCs.

REVIEWERS' COMMENTS

Reviewer #1 (Remarks to the Author):

The manuscript submission by Dallmeier and colleagues reports on pre-clinical assessment of their yellow fever virus-based COVID-19 vaccine in the hamster model. The authors demonstrate that 18 amino acid changes of the prototypic spike sequence expressed by their vaccine backbone broadens protective antibody responses against multiple lineages. The findings of the authors support updating of the licensed COVID-19 vaccines, and are in agreement with recently authorized clinical use of bivalent COVID-19 vaccines. A major strength of their vaccination approach is likely induction of mucosal immunity against SARS-CoV-2 infection.

A major weakness of the experimental approach is the abbreviated sequential vaccination schedule with 7-day interval and challenge infection at 21 days post vaccination. This abbreviated vaccination schedule does not allow establishment of memory responses following prime immunization or recall of memory responses.

We convene with the reviewer that future studies with a focus on longevity and boostability of immunity need to explore extended schedules. As explained in the revised manuscript and our previous rebuttal, the day 0+7 schedule was employed following a previously validated rapid immunization scheme (lines 110-111); not claiming any optimization.

Our comparative study did not primarily aim at comparing immunization schedules or memory responses, yet rather the profiling of two vaccine antigens regarding their variant coverage. In fact, as demonstrated before, YF-S0 yielded already high seroconversion rates and protection from single-dose vaccination (Sanchez-Felipe *et al.* Nature 2021; Ma *et al.* EBioMedicine 2022). The latter was confirmed again in the current study for YF-S0* (new Figure 3), whereby a second dose at day 7 helped to close remaining gaps in immunization and reduced the chance of primary vaccination failure (lines 201-204). Similar was already observed in our assessment of prototypic YF-S0; in particular when 10-fold lower vaccine doses were compared head-to-head following a single and dual immunization schedule, clearly establishing a dose-dependency of immunization (Sanchez-Felipe *et al.* Nature 2021).

In the extended M&M section we included an extra narrative for justification, also mentioning possible limitations of the thus chosen immunization schedule (lines 432-438). There, we appreciate that a full development of B cell responses prior to the second dose can not be expected following such a rapid two-dose immunization schedule. Nevertheless, it may have at least resulted in a more consistent immunization. Finally, despite probably suboptimal, two doses given 7 days apart induced high levels of nAb and resulted in protection against SARS-CoV-2 challenge in macaques (Sanchez-Felipe *et al.* Nature 2021).

In general, the manuscript requires proofreading to improve clarity in some parts of the text. Do the 2nd and 3rd paragraphs of Vaccination and challenge, (1) standard setting, provide redundant information?

The respective M&M section has been carefully reassessed. Both paragraphs describe our basic or standard setting (1), with two almost identical, hence seemingly redundant vaccination-challenge regimens; appreciating minor variations on the theme. YF-S0: vaccination d0+7, bleed and infection d21 (using prototype, VOCs Alpha and Beta); YF-S0*: vaccination d0+7, bleed d21 and infection d24 (using VOCs Alpha, Beta, Gamma and Delta).

The authors should revise the third sentence or the Abstract to indicate their yellow fever virus vaccine platform.

Done as requested.

Reviewer #2 (Remarks to the Author):

The revised manuscript by Sharma et al. was revised in response to the comments provided. Most of my concerns have been addressed in the revision and the new added data shown in Figure 3 strengthen the overall message that the YF-S0* suppressed YF-S0 vaccine and that the first generation vaccines should be updated against emerging VOCs.

We thank the Reviewer for the appreciation.